# Image Inpainting Forgery Detection: A Review

**DOI:** 10.3390/jimaging10020042

**Published:** 2024-02-02

**Authors:** Adrian-Alin Barglazan, Remus Brad, Constantin Constantinescu

**Affiliations:** Faculty of Engineering, Computer Science, “Lucian Blaga” University of Sibiu, 550024 Sibiu, Romania

**Keywords:** image inpainting, object removal detection, forensic forgery

## Abstract

In recent years, significant advancements in the field of machine learning have influenced the domain of image restoration. While these technological advancements present prospects for improving the quality of images, they also present difficulties, particularly the proliferation of manipulated or counterfeit multimedia information on the internet. The objective of this paper is to provide a comprehensive review of existing inpainting algorithms and forgery detections, with a specific emphasis on techniques that are designed for the purpose of removing objects from digital images. In this study, we will examine various techniques encompassing conventional texture synthesis methods as well as those based on neural networks. Furthermore, we will present the artifacts frequently introduced by the inpainting procedure and assess the state-of-the-art technology for detecting such modifications. Lastly, we shall look at the available datasets and how the methods compare with each other. Having covered all the above, the outcome of this study is to provide a comprehensive perspective on the abilities and constraints of detecting object removal via the inpainting procedure in images.

## 1. Introduction

With the improvements and innovations in the last few years in the fields of image inpainting techniques and machine learning, an increasing amount of altered and fake media content has invaded the internet. In the current paper, a thorough review of the current inpainting mechanism is undertaken, and the current state-of-the-art techniques to detect these alterations are analyzed.

Nowadays, in our modern society, we rely on technology. This can be seen as an advantage but also as a drawback. The evolution of technology has impacted the lives of each of us. We are just a click away from an almost infinite amount of information that can be accessed at any time. Most of the time, people rely completely on the information that they find online and form their opinions based on these facts, but unfortunately, this is not always a safe approach. The information that can be found online can sometimes be distorted or even false. This is the reason its accuracy needs to always be checked. We tend to believe that false information can be transmitted only through textual content, but this is not entirely the case. Nowadays, images and videos are also tools for transmitting information. We use them daily, and we became accustomed to believing everything we see to be the truth. The most powerful example, in this case, are the images and videos that we see and upload on social networks. This example shows that it is equally important to check the authenticity of images as it is to check the trustworthiness of written text. All the reasons stated above imply that there is a great need to detect forgeries in images and videos.

The science area focusing on tampered image and video detection is called media forgery detection. The area is quite vast and has had increasing interest as described in recent bibliometric studies conducted by [1,2,3] (Figure 1: Number of forgery detections per paper in recent years). Forgery detection methods can be divided into two categories: active and passive. For active methods, the focus is embedding some metadata onto images at the time of creation, and these can later be used to validate the authenticity of images. On the other hand, passive methods, which are sometimes also called blind methods, do not offer much specific information, and thus one must rely entirely on the possible artifacts introduced in the tampering process.

If one looks at the tampering process according to the father of digital image forensics, Hany Farid, as mentioned in [4], forgery detection can be performed in the following ways:Active methods—briefly, the main idea here is to incorporate various information that can be validated later at the moment of image acquisition or at the moment an image undergoes certain operations like resize, crop, etc.Passive methods—here, the scope is quite big. Some of these methods focus on the peculiarities of image capturing, camera identification, noise detection, and image inconsistencies or on specific types of traces, which are usually introduced by a forgery mechanism; e.g., for a copy-paste forgery (combining information from multiple images), some traces like inconsistent coloring, noising, blur, etc. might be noticed.

A more detailed schema based on the above-mentioned categories can be seen in Figure 2, using the comprehensive work by Pawel Korus in his research paper [5]. Based on our analysis, the categorization is structured a bit differently. The first difference is related to camera traces, where all steps/artifacts are grouped together. The reason for this grouping is that each of the sub-methods under camera that can be analyzed in depth has to be undertaken in the context of several types of cameras, as one cannot generalize trace artifacts; e.g., regarding camera lens traces, although there are predictable patterns like barrel or pincushion, the specific degree and characteristics of radial lens distortion can vary from one camera to another. This categorization is important because camera traces can be used to determine the forged area. As can be noticed in later chapters, the authors initially tried to focus only on one type of artifact, but recent studies suggest that the best approach would be an ensembled method. Let us consider that all types of image operations that are conducted on a source image without blending information from other images would fall under this category. Therefore, the new category contains items like object removal, resampling, blurring, gamma correction, etc. This is particularly important because the category is redefined as an operation on an image, solely based on the statistical data of the image itself; e.g., when inpainting an image, data is filled based on the overall statistical analysis of the image.

Looking from the “attacking” point of view, for the passive methods, other classifications can be observed from the traces that forgery methods might introduce, as shown below:Image composition and enhancement methods—in these cases, the picture is altered, for example, by copying parts of the original image into the same image. Of course, steps like re-sampling and rescaling can be included; usually, both resampling and rescaling are not methods of altering at their core but are used as a step to apply copy-paste or splicing methods.Splicing—the forged media is obtained by combining several images into one by, e.g., taking a picture of someone and adding it within another one.

The current paper conducts a deep evaluation of the current methods for detecting inpainting/object removal inside images. The material is divided into the following parts. The first part provides a review of the current state-of-the-art methods in inpainting methods, with a deep focus on object removal. Artifacts are introduced based on each category of inpainting methods. Going further, an analysis is performed with all the pros and cons of each method, the dataset is reviewed, the way they behave in real-world scenarios is verified, and they are compared to others in terms of quality. After this thorough review, the focus is shifted to describe the detection of inpainting problems as a general matter. After that, this paper analyzes forgery detection methods with a focus on object removal detection. Some older variants are presented first, and their main ideas and their performance are assessed. The study continues by also analyzing other relevant forgery detection mechanisms and investigating if they can be applied successfully to object removal tasks. For each relevant method, a comprehensive analysis is undertaken on the pros and cons alongside an analysis of how they work outside the tested datasets. Furthermore, an additional analysis is performed on available datasets used for the evaluation of these methods and the issues they raise. Lastly, all the relevant findings are briefly summarized, and the relevant areas of improvement are also diligently addressed.

## 2. Inpainting Methods

Image inpainting is sometimes called an inverse problem, and usually these types of problems are ill-posed. The problem of inpainting consists in finding the best approximation to fill in the region inside the source image and comparing it with the ground truth. All the algorithms that tackle this problem begin with the assumption that there must be some correlation between the pixels present inside the image, either from a statistical or from a geometrical perspective. The objective of inpainting is to minimize the difference between the original image, I, and the reconstructed image, R, in the domain, D. This is typically achieved by defining an appropriate loss function that quantifies the differences between I and R in region D. In other words, the inpainting problem can be formulated as an optimization problem that minimizes an objective function. The objective function typically consists of two main components: a term that penalizes the deviation of the inpainted values from the known values in the known region, D; and a term that penalizes large variations in the inpainted values, encouraging smoothness and preventing overfitting. Thus, the first operation ensures that the inpainted image is consistent with the available information, and the second helps to preserve the natural structure and appearance of the original image. The inpainting methods can be split into the following three categories:Diffusion-based or sometimes called partial differential equations-based (here, we are also going to include TV methods as well);Exemplar-based or patch-based, as referred to in some other papers;Machine learning is undertaken irrespective of their model architecture.

The following section focuses on describing the most cited and recent state-of-the-art methods to better understand, the overall artifacts introduced in the inpainting procedure.

### 2.1. Diffusion-Based Methods

The term diffusion (from a chemistry point of view) is an action in which items inside a region of higher concentration tend to move to a lower concentration area. In the analysis undertaken in [6], in which the inpainting process is inspired by the “real” inpainting of canvas, the process consists of the following steps:Global image properties enforce how to fill in the missing area;The layout of the area, δD, is continued into D (all edges are preserved);The area, *D*, is split into regions, and each region is filled with the color matching δD (the color information is preserved from the bounding area, δD, into the rest of the D area);Texture is added.

The first step in all the inpainting algorithms is to apply some sort of regularization. It can be isotropic, with some poor results, anisotropic, or any other type of regularization. This is performed to ensure that image noise is removed, so that it shall not interfere in the computation of the structural data needed in the next step.

To apply diffusion, the structural and statistical data of the low-level image must be identified. Based on this data, if on an edge in the *δD* area, the algorithm must conserve the edge identified, and if the *δD* area belongs to a consistent area, the algorithm can then easily replicate the same pixel information from the border. To retrieve image geometry, one can use isophotes, which are curved on-surface connecting points of the same values. For this, one needs to first compute the gradient on each point in the margin area and then compute the direction as normal in relation to the discretized gradient vector.

Having performed these steps, the initial algorithm from [6] is just a succession of anisotropic filtering, followed by inpainting, and then, this is repeated several times. Later, the authors in [7] proposed an improved version of their initial algorithm. This idea was inspired from the mathematical equations of fluid dynamics, specifically the Navier–Stokes equations, which describe the motion of fluid. The proposal was to use the continuity and momentum equations of fluid dynamics to propagate information from known areas of the image towards the missing or corrupted areas. This was an improved version of the higher PDE version presented initially. As a follow-up of his original work, Bertalmio proposed the use of third-order PDE in [8], which is a better continuation of edges. At the same time, Chan and Shen developed similar algorithms [9,10], in which they postulated the use of the local curvature of an image to guide the reconstruction of missing or obscured parts. Using Euler’s elastica model, they could predict what the missing parts of an image might look like. Both Euler’s elastica and PDE-based inpainting are effective methods for image inpainting and have their own advantages and disadvantages. Euler’s elastica is particularly well-suited for images that contain thin, flexible objects, while PDE-based inpainting is well-suited for images that are smooth and locally consistent. Depending on the specific characteristics of the image and the desired outcome, one method may be more appropriate than the other. In recent year, the focus for diffusion-based inpainting has moved towards increasingly complex PDE forms; e.g., in [11] using high-order variational models is suggested, like low curvature image simplifiers or the Cahn–Hilliard equation. Another recent paper that goes in the same direction is [12], which integrates the geometric features of an image—namely the Gauss curvature. Still, even these methods introduce the blurring artifact also found in the initial papers [6,7]. To surpass these challenges in the current models, with second-order diffusion-based models that are prone to staircase effects and connectivity issues and fourth-order models that tend to exhibit speckle artifacts, a newer set of models must be developed. The authors Sridevi and Srinivas Kumar proposed several robust image inpainting models that employ fractional-order nonlinear diffusion, steered by difference curvature in their papers [13,14,15]. In their most recent paper [16], a fractional-order variational model was added to mitigate noise and blur effectively. A variation of DFT is used to consider pixel values from the whole image, and this is not by relying strictly on only the neighboring pixels.

Another method that yields better results than standard PDE is the use of total variation inpainting. The core idea is to minimize the total variation of the inpainted image, effectively reducing the abrupt changes in intensity. TV inpainting incorporates a regularization term into the optimization problem. This term typically comes in two types: L1 and L2 regularization. L1 regularization encourages sparsity in the gradients of the inpainted image, promoting piecewise constant solutions with sharp edges. On the other hand, L2 regularization results in smoother images with gradual transitions. The choice between these regularization terms determines the trade-off between smoothness and fidelity in the inpainted result. First-order TV inpainting focuses on minimizing the L1-norm of image gradients and is effective for restoring image edges and preserving fine structures. Minimizing the L1-norm of gradients encourages sparsity in the gradient domain, leading to piecewise constant solutions. One of the key advantages of first-order TV inpainting is its applicability in various scenarios, including image denoising and deblurring. This method excels in reconstructing images with well-defined edges and minimal noise. Second-order TV inpainting extends the TV concept by considering gradients of gradients, also known as the Laplacian operator. This extension allows for the preservation of more complex image structures, including textures and patterns. By considering higher-order derivatives, second-order TV inpainting can produce inpainted images with improved fidelity. However, second-order TV inpainting introduces computational challenges due to the increased complexity of the optimization problem such as in [17] where the split Bregman technique is used.

Based on the multitude of research on inpainting using PDE or TV methods, there are still some areas to be improved, and some artifacts are easily identifiable. In the following section, two methods are analyzed in terms of the artifacts introduced in the inpainting process. The first is based on the Cahn–Hilliard equation [11] and is a fourth-order PDE that describes the phase separation process in materials and has been adapted for inpainting tasks in image processing, while the second analyzed method is based on total variation [17] and uses a combined first- and second-order total variation. The first analyzed sample is presented in Figure 3: on the left side is the original image [11], and on the right is the mask applied to that image. The mask created for the PDE method is created specifically for this test, and it tries to simulate a scratch on the image. The inpainting methods will try to generate pixels that are as accurate as possible to fill in the white line.

In the below image, Figure 4, the result of the Cahn–Hilliard inpainting method can be observed. The parameters used to generate the image are the standard suggested by the authors (max iteration = 4000, epsilon = [100,1], lambda = 10).

The first artifact that can easily be distinguished is the discontinuity; e.g., when transitioning from the maya blue region to the green and yellow region. PDE methods can sometimes introduce noticeable discontinuities or abrupt transitions between inpainted regions and the surrounding image. Another noticeable artifact is the incorrect color matching as shown in Figure 5: in cases where color is not properly preserved or matched during inpainting, the inpainted regions may not blend seamlessly with the rest of the image, leading to visible differences in color or texture. This can be easily noticed in the canary color region. Another visible artifact introduced during the inpainting method is blurring because diffusion-based PDEs tend to smooth the image as they propagate information, which can lead to a loss of sharpness and detail.

In the above scenario, the TV variation [17] is able to correctly reconstruct pixels, at least from a human perspective. However, upon examining the pixel values with the area values, it becomes apparent that the reconstruction is not flawless, as there are minor discrepancies in colors, as shown in Figure 6.

The second analyzed scenario is more complex, and we are going to focus on the TV method only because the classic PDE method usually struggles with larger/highly texturized areas. In Figure 7, on the left side is the original image [17], and on the right is the mask applied to that image. The inpainting methods will try to generate pixels that are as accurate as possible to fill in the white line. In the below image, the result of the TV inpainting method can be observed. The parameters used to generate the image are the standard one suggested by the authors (maxIteration = 100, Gamma1 = 5, Gamma2 = 8).

Some of the above already discussed artifacts from the classic PDE method also appear in the context of the TV methods as well when the inpainting region is bigger and more complex: blurring around the area, over smoothing inside the inpainted area, and inaccurate texture matching/color matching of the water. In addition to the above artifacts, some other interesting artifacts are exhibited after using the TV methods: staircase artifacts and halos. Staircase artifacts, also known as “blocking artifacts,” are a type of visual distortion that occurs in images when there is a sharp transition or abrupt change between adjacent blocks or regions, and this can be seen in the below sample. These artifacts are characterized by the appearance of visible grid-like patterns resembling a staircase along the edges or boundaries of these blocks. This type of artifact can be observed in the zoomed area in the below picture (Figure 8).

TV inpainting methods can also introduce halos, which are bright or dark rings that appear around the edges of the inpainted region. This is more likely to occur when the inpainted region has a lot of contrast between the inpainted region and the surrounding area.

Summarizing the diffusion inpainting methods, it is found that they usually rely on second or higher order partial derivatives or the total variation of energy to be able to “guess” the missing area. One of the major drawbacks of these methods is as follows: depending on the complexity of the inpainted region, they might introduce blurring or some sort of color and texture inaccuracy, which in turn will affect the entire image. The blurring effect usually happens because of either the regularization operation or due to the diffusion process. Halos associated with TV are another cause of the tendency to over smooth edges. This over smoothing occurs because the TV minimization objective encourages the image to have constant brightness across each pixel, which can lead to abrupt changes in brightness at the edges. These abrupt changes in brightness appear as halos around the inpainted region. Due to this blurring effect, in theory, image inpainting via PDE and TV can be detected via some sort of inconsistency in the blurring effects of various regions or inconsistencies inside the color/texture, but this is harder to detect without a priori knowledge of the original image.

### 2.2. Exemplar-Based Methods

At the same time, a newer approach based on texture synthesis started to gain more momentum. The main inspiration came from [18] in which A. A. Efros and T. K. Leung introduced a non-parametric method for texture synthesis, where the algorithm generates new texture images by sampling and matching pixels from a given input texture based on their neighborhood pixels, thus effectively synthesizing textures that closely resemble the input sample. This approach was notable for its simplicity and ability to produce high-quality results, making it a foundational work in texture synthesis. The primary goal of this approach was to enhance the reconstruction of the image section area that is missing. However, the challenges brought by texture synthesis are slightly different from those presented by classic image inpainting. The fundamental objective of texture synthesis is to generate a larger texture that closely resembles a given sample in terms of visual appearance. This challenge is also commonly referred to as sample-based texture synthesis. A considerable amount of research has been conducted in the field of texture synthesis, employing strategies such as local region growing or holistic optimization. One of the main papers that gained a lot of attention was the work of [19]. In this paper, Criminisi presented a novel algorithm for the removal of large objects from digital images. This technique is known as exemplar-based image inpainting. This method is based on the idea of priority computation for the fill front, and the use of the best exemplar selection for texture synthesis. Given a target region, Ω, to be inpainted, the algorithm determines the fill order based on the priority function *P*(*p*), defined for each pixel, p, on the fill front: *∂*Ω. *P*(*p*) = *C*(*p*) * *D*(*p*), where *C*(*p*) is the confidence term, an indication of the amount of reliable information around pixel p; *D*(*p*) is the data term, a measure of the strength of the isophotes hitting the front at p. The algorithm proceeds in a greedy manner, filling in the region of highest priority first with the best match from the source region, Φ. This is identified using the sum of squared differences (SSD) between patches. The novel aspect of this method is that it combines structure propagation and texture synthesis into one framework, aiming to preserve the structure of the image while simultaneously considering the texture. It has been demonstrated to outperform traditional texture synthesis methods in many complex scenes, and it has been influential in the field of image processing. 

In recent years, the methods have become increasingly complex and try to exploit various artifacts inside images and analyze more in-depth the structure near the area to be inpainted. Approaches like [20] utilize a patch-based approach that searches for well-matched patches in the texture component using a Markov random field (MRF). Jin and Ye [21] proposed an alternative patch-based method that incorporates an annihilation property filter and a low-rank structured matrix. Their approach aims to remove an object from an image by selecting the target object and restricting the search process to the surrounding background. Additionally, Kawai [22] presented an approach for object removal in images by employing a target object selection technique and confining the search area to the background. Authors have also explored patch-based methods for recovering corrupted blocks in images using two-stage low-rank approximation [23] and gradient-based low-rank approximation [24]. Another sub-area of focus for some authors was representing information by first “translating” the image into another format, the so-called sparse representation like DCT, DFT, DWT, etc. It is relevant to mention a few interesting research papers [25,26]. They obtained excellent quality while maintaining the uniformity of the area to be inpainted, but if the area is at the edge of various textures, the methods introduce some pretty ugly artifacts that make the methods unusable.

Various authors like [27,28] have suggested that another classification of the inpainting procedure can be undertaken. They either suggest adding a subdivision based on the sparse representation of images (like authors have suggested in [25,29,30,31]) and then later trying to apply existing algorithms to this representation, or a so-called mixed/fusion mode in which authors try to incorporate ideas from both worlds: from diffusion-based and from texture synthesis (patch copying). In the latter category, some ideas are worth noticing like the one Bertalmio explored in his [32] study, in which they combined a PDE-based solution together with patch synthesis and a coherence map. The resulting energy function is a combination of the three metrics. A similar idea to Bertalmio in the above mentioned work is the research by Aujol J, Ldjal S, and Masnou S in their [33] article, in which they try to use exemplar-based methods to reconstruct local features like edges. Another investigation of the same idea was the work of Wallace Casaca, Maurílio Boaventura, Marcos Proença de Almeida, and Luis Gustavo Nonato in [34], in which they combine anisotropic diffusion with a transport equation to produce better results. Their suggested approach of using a cartoon-driven filling sequence has proven to be highly effective for image inpainting using both PSNR and speed as metrics.

Summarizing the exemplar (or patch-based) inpainting methods, they try to undertake the following three steps:find the best order to fill the missing area;find the best patch that approximates that area;try to apply, if needed, some processing on the copied patch in order to ensure that both the local and global characteristics are maintained.

Similar to the diffusion-based methods, the artifacts introduced in the inpainting process by patch-based methods are analyzed. The first selected method is that used in the initial Criminisi paper [19], while the second is a more recent patch-based approach [35]. While patch-based inpainting can produce visually plausible results, the methods are susceptible to various artifacts, such as staircasing, blurriness, and mismatched textures. The selected images are presented below together with the mask applied to them. In Figure 9 and Figure 10, we can observe the original, the mask applied, and the overall results using the methods from Criminisi and from Huang. In Figure 11, the two results are presented side by side.

The first artifact analyzed is incorrect color or texture continuation. It arises when the inpainting algorithm fails to accurately reproduce the color or texture of the surrounding area when filling in the missing region or when the inpainting algorithm copies patches that do not fit well with the surrounding structure of the image. This can lead to unnatural and unrealistic transitions between the inpainted region and the surrounding pixels. One of the primary reasons for incorrect color or texture continuation is the limited information available to the inpainting algorithm. When copying patches from the surrounding area, the algorithm only considers the colors and textures within the patch itself. This can be insufficient to accurately determine the continuation of color or texture across the missing region, especially in areas with complex or subtle variations. Another factor contributing to this artifact is the randomness involved in the patch selection process. When choosing patches from the surrounding area, the algorithm does not necessarily select patches that are most similar in color or texture to the missing region. This can lead to discrepancies in the color or texture representation, making the transition between the inpainted region and the surrounding pixels appear unnatural. This type of artifact is presented in the zoomed image in Figure 12 after applying the Criminisi method:

Another type of common artifact is aliasing, also known as the staircase effect or jaggy edges. It is a common artifact that can occur in patch-based image inpainting. It arises due to the limitations of the algorithm in accurately replicating the delicate details of the original image, particularly along high-frequency edges. When the inpainting algorithm copies patches from the surrounding area to fill in the missing region, it often struggles to maintain the sharp transitions and gradients that define edges. This is because the algorithm is averaging the pixel values within the patch, which can result in a smoothing effect that blurs the edges. Therefore, the reconstructed edges appear jagged or pixelated, resembling a staircase-like pattern. This is particularly evident in regions with high contrast and sharp transitions, such as the outlines of objects or the edges of textures. The aliasing effect is caused by the sampling process employed by the inpainting algorithm. When copying patches from the surrounding area, the algorithm relies on a fixed sampling rate, which means that it only considers a limited number of pixels from each patch. This limited sampling cannot fully capture the fine details of the image, especially along the edges where the pixel values vary rapidly. As a result, the algorithm struggles to accurately reproduce these details when filling in the missing region, leading to the staircase effect. This is visible in Figure 13.

Blurring is another common artifact that can occur in patch-based image inpainting, particularly when dealing with intricate textures or high-contrast edges. It arises due to the averaging process employed by the inpainting algorithm when copying patches from the surrounding area. When filling in the missing region, the algorithm averages the pixel values within the copied patch, effectively smoothing out the fine details and reducing the sharpness of edges. This blurring effect becomes more pronounced in regions with complex textures or sharp transitions, as the averaging process fails to capture the intricate patterns and gradients. The blurring artifact primarily stems from the limited receptive field of the inpainting algorithm. When copying a patch, the algorithm only considers the pixels within that patch, essentially operating within a narrow area. This restricted view prevents the algorithm from fully comprehending the wider context of the image, including the intricate details and sharp edges that define textures and objects. For example, by applying this method [35], in Figure 14, the zoomed area is visualized.

The blurring effect is more pronounced, particularly when the region to be inpainted has a high level of texture. For example, in Figure 15, the Huang method is applied.

Since the region being inpainted is textured, the outcome is highly blurred, and in the below zoomed region, this can be observed in Figure 16.

By analyzing the artifacts these methods introduce, the methods can be categorized into two groups: methods that simply “copy-paste” unconnected/unrelated regions (patches) into the missing area and methods that do some enhancement/adaptation of the patch values. The first category is straightforward to determine via a forensic algorithm: it relies solely on the fact that a given region (usually several times greater than the patch used for inpainting) is comprised of patches that have “references” in other areas. The main problems here are how to determine the correct patch size to be able to determine “copies” and speed and last but not least, how to correctly eliminate false positives (especially when the patch size is smaller, and the window step is small as well). The second category of inpainting patch-based methods that do not simply copy the source patch to the destination is a little harder to detect. The above algorithm, where similar patches are searched for, can no longer be applied, and thus, some new heuristic must be introduced and evaluation on how much resemblance the patches have should be revisited. If the patch similarity mechanism is updated, the problem of false positives increases exponentially and on images with large smooth textures, a lot of false positives might be reported. Still, there are some clues that can be used as starting points, like incorrect texture inconsistencies or blurring.

### 2.3. Machine Learning-Based Methods

Deep learning methods have significantly improved the accuracy and effectiveness of image inpainting. Given its exceptional outcomes, numerous researchers are motivated to revise, employ, or even adopt novel methodologies. Image inpainting includes deep learning techniques, specifically convolutional neural networks (CNNs). The objective is to utilize a training set of images to instruct the convolutional neural networks (CNNs) in the task of filling or completing the sections or regions that are absent in the images. The initial approach to address image inpainting is training a specialized model to fill in a missing area at a particular position inside an image. Other architecture like encoder-decoder, combined with generative adversarial networks and utilizing deep learning techniques, have outperformed previous methods. Based on this observation, some authors have tried to categorize machine learning inpainting methods into CNN- and GAN-based methods [36]. In [37], a comprehensive overview of the latest prominent image inpainting techniques is provided based on deep learning in two groups: single-stage methods and progressive methods. Other researchers categorized deep learning-based methods into three groups: those utilizing autoencoders, those employing generative models, and those focusing on network structure optimization. Prior studies failed to account for methods using diffusion-based models or transformers. For a detailed review on machine learning-based techniques, authors in [37,38] offer particularly good overviews of the current state-of-the-art methods. Recent years have also shown promising results in models based on diffusion or more complex models based on transformers. Grouping inpainting methods based on model structure is a practical and insightful approach for several reasons, especially in inpainting detection, as listed below:-Underlying Mechanism and Approach—Different model structures use fundamentally different mechanisms and approaches to inpainting. For example, autoencoders leverage an encode-decode framework, GANs rely on adversarial training between a generator and discriminator, diffusion models use a probabilistic approach to gradually construct an image, and transformers process global dependencies in data. Understanding these core mechanisms helps in selecting the right model for a specific type of inpainting task.-Strengths and Limitations—Each model structure has its unique strengths and limitations. Autoencoders are efficient but may struggle with high-frequency details, GANs can generate high-quality images but often face training stability issues, diffusion models excel in producing coherent and detailed images but are computationally intensive, and transformers are excellent at capturing long-range dependencies but may face challenges with local coherence. Grouping by structure allows for a clear comparison of these strengths and limitations.-Artifacts and Quality of Output—Probably, from a detection point of view, the most important aspect is that similar artifacts are introduced by the same type of model. Different structures tend to produce different kinds of artifacts in the inpainted images, as discussed earlier. By grouping methods based on structure, it is easier to anticipate the types of artifacts that might arise and choose a method that minimizes undesired effects for a given application.

Therefore, the methods for inpainting are categorized as follows:-GAN-based methods;-Diffusion models;-Transformers.

In the below sections, only relevant methods from each group are analyzed, along with relevant artifacts from each group. In the area of machine learning, the most referenced method, which is attributed to Deepak Pathak’s paper [39], suggests the use of a system composed of an encoder and a decoder. The encoder focuses on retaining data information (extracting it), and the decoder’s responsibility is to generate features based on the encoder’s learned data. Starting with this approach, several methods have initially been suggested, like FCN that has two neural networks with skip connections between them. An improved version of this FCN version is U-Net architecture, which resembles FCN but uses summation as a skip connection mechanism and employs concatenation. The advantage of using concatenation is that it can retain more detailed data. One of the existing problems with inpainting is how to generate the missing area in a highly texturized area. To address this challenge, some authors have proposed different mechanisms to exploit both global and local texture information. Another point of variation between inpainting methods is the convolutions used. The focus is more on the convolution applied at the decoder level (deconvolution or as some authors call it, transposed convolutional layer) because it is the one responsible for introducing several types of artifacts. From the analysis, the most used convolutions are simple (or standard) convolution—which is good for reconstruction, especially for rectangular-based shapes—and gated convolution, the main idea of which is to be able to fill in irregular holes by using a mask that is updated after each convolution [30,37,38].

Recent research with relevant satisfactory results are [40,41,42]. The methods used rely on Fourier convolutions and perceptual loss. Their results are impressive in the CelebA [43] and places [44] datasets. An improvement on the Lama model was presented by the authors in [45] on the learned perceptual image patch similarity (LPIPS) metric. Their method starts with a noisy image and applies denoising, filling in the missing data based on the known area. In [46], the authors suggested another approach in that they apply an already established classic method of inpainting from OpenCV (Telea and NS methods) and use a CNN model to learn the reconstruction of these features. As a backbone, they use a VGG16 model and as features, the authors used three distinct traits: image size, RGB information, and brightness levels. The results are straightforward when the area to be inpainted is uniform (e.g., the middle of the sky), but when the region to be generated is at the border of several highly texturized areas, the methods do not yield proper results. Recently in [47], they suggested that the task of inpainting should be divided into two separate stages. In the first stage, they used two separate DRGAN modules—one to generate the content of the inpainted area and one to generate the edges of the missing area; they generated a label image where 1 is the edge, and 0 represents the background. This information is crucial in the second stage of the algorithm, where the authors used a fine-grained network to generate more coarse pixel information based on the label edges and the already generated data. Again, they used a DRGAN architecture, a deep residual generative adversarial network, for this part of the method. Analyzing the results and the comparison with some state-of-the-art methods, the proposed method can reconstruct highly texturized areas but has some limitations in what the authors call “overly complex” regions. Authors in [48] diverged from the mainstream usage of transformers and incorporated a discrete wavelet transformer along the convolutional layers. Still, for up sampling, the authors used the standard transpose convolution, which generates checkboard artifacts. Another approach with particularly good results is [49], in which the authors combined autoencoders and transformers on a Resnet architecture. The idea behind using transformers is that they can better represent details and thus be able to reconstruct the missing area, but the authors still used the same type of architecture (Resnet), which employs the same type of up sampler. In [50], the authors present a new class of diffusion models called latent diffusion models (LDMs) that achieve state-of-the-art performance for image inpainting and other tasks, including unconditional image generation, semantic scene synthesis, and super resolution. LDMs are based on the idea of decomposing the image formation process into a sequential application of denoising autoencoders. This allows LDMs to be trained with a more efficient training process, and it also allows for more flexible control over the image generation process. In the paper, the authors compare LDMs to pixel-based diffusion models (PDMs), which are the most common type of diffusion model. They found that LDMs are significantly more efficient than PDMs, and they also produce higher quality results. The authors conclude that LDMs are a promising new approach to image synthesis, and they believe that they have the potential to revolutionize the way that images are generated and used. Among the limitations, the most common ones are mode collapse; in some cases, LDMs may exhibit mode collapse where they consistently produce similar-looking inpainted images, even when presented with different input masks and texture artifacts. LDMs can sometimes introduce texture artifacts, such as blurry or unnatural patterns, in the inpainted regions. These artifacts arise from the model’s tendency to smooth out sharp edges and fine details in the input image.

Diffusion-, GAN-, and transformer-based image inpainting methods each have their own strengths and weaknesses in terms of artifact production. Here is a summary of the common artifacts associated with each method.-Diffusion-based inpainting▪Texture artifacts: Diffusion-based models can introduce blurry or unnatural textures in the inpainted regions, especially when dealing with complex textures or high-resolution images. This is because diffusion models gradually reduce noise from the input image, potentially over smoothing fine details.▪Color inconsistencies: Color inconsistencies can also occur with diffusion-based inpainting, leading to discrepancies in color saturation or hue between the inpainted areas and the surrounding pixels. This can make the inpainted image appear unnatural or unrealistic.▪Ghosting artifacts: Diffusion-based models may introduce ghosting artifacts around the edges of the inpainted areas, making them look detached from the surrounding image. This can be caused by the model’s tendency to overemphasize the edges of the missing regions.-GAN-based inpainting▪Mode collapse: GAN-based models can sometimes suffer from mode collapse, where they consistently produce similar-looking inpainted images even when presented with different input masks. This can limit the diversity and creativity of the generated inpainting results.▪Fake artifacts: GAN-based models may introduce artifacts that appear fake or artificial, such as checkerboard patterns, blurry textures, or unnatural patterns. This can happen when the model struggles to capture the fine details and subtle textures of the original image.▪Color artifacts: Color artifacts can also occur with GAN-based inpainting, especially in early generations or when the model is not trained properly. This can make the inpainted image appear unnatural or unrealistic.-Transformer-based inpainting▪Pixelation: Transformer-based models can sometimes produce pixelated artifacts, especially when inputting low-resolution images or when generating images with sharp edges or high contrast. This is because the attention mechanism used in transformers may focus on a small number of pixels, leading to a loss of detail in the final output.▪Checkerboard patterns: Checkerboard patterns can also be introduced by transformers, especially when generating images with sharp edges or high contrast. This is because the attention mechanism may not be able to smoothly transition between different regions of the image, leading to a checkerboard-like appearance.▪Color banding: Transformers can also introduce color banding artifacts, which appear as horizontal or vertical stripes of color. This is typically caused by the model’s inability to accurately represent smooth gradients of color.

In general, diffusion-based inpainting is known for its ability to produce smooth and realistic results, but it may introduce texture artifacts and ghosting artifacts. GAN-based inpainting can generate diverse and realistic inpainting results, but it is more prone to mode collapse and fake artifacts. Transformer-based inpainting excels at high-resolution inpainting but may suffer from pixelation and checkerboard artifacts.

Machine learning-based inpainting methods, particularly those using convolutional neural networks (CNNs) like Lama (large mask inpainting), have made significant advances in image restoration and object removal. However, these methods still introduce specific artifacts, which can vary depending on the algorithm’s design, training data, and the complexity of the inpainting task. The most common artifacts introduced by these methods include blurring, inconsistent textures, color discrepancies or repetition, and pattern duplication. Blurring, loss of detail, and over smoothing: machine learning algorithms, especially earlier or simpler models, can struggle to reproduce fine details. This often results in inpainted areas that are blurrier or less detailed compared to the original image’s parts. Sometimes, these models may also smooth out the natural texture and structure of the image, making the inpainted region appear artificial, like in Figure 17:

Inconsistent textures arise when the inpainting process duplicates patches of pixels from the surrounding area that do not correspond to the texture of the area that is missing. This might result in an artificial appearance, giving the impression that the area that is missing has been replaced with a dissimilar substance. For instance, in the Figure 18 example, if the inpainting algorithm removes the knife, the outcome appears artificial due to the mismatch in textures between the removed area and the rest of the image.

In the above image, color disparities can be seen as well. These artifacts arise when the inpainting algorithm employs hues that do not correspond to the hues of the adjacent region. This can result in an artificial appearance, giving the impression that the area that is missing has been replaced with a distinct object.

Edge artifacts manifest when the inpainting method is unable to precisely rebuild the boundaries of the absent area. These imperfections can result in the appearance of uneven or blocky edges that can be seen in the restored image. For instance, if an inpainting method is employed to eliminate an object from a forest image (see Figure 19), the edges of the object would remain discernible in the inpainted image due to the system’s inability to precisely rebuild the intricate forms of the leaves and branches.

Structural and geometric errors arise when the inpainting process fails to accurately restore an image’s general structure and geometry. These actions may result in the formation of discontinuities or boundaries between the restored area and the adjacent pixels as well as the potential alteration of the image’s general morphology, as can be seen in the above example.

Although less common than in patch-based methods, some ML algorithms can produce repetitive patterns, especially when dealing with large inpainting areas. For example, sometimes the Lama method generates patterns like the ones presented in Figure 20.

From a detection point of view, these methods are becoming more challenging due to their ability to propagate patches that are indistinguishable from the rest of the image. Also, due to their nature to complete large areas, they can reconstruct entire image characteristics. Various methods have been proposed as attack vectors, but they are focusing on the artifacts introduced by the various up sampling steps. One can focus on artifacts like the ones described above, but they are harder to implement because there is the need for complete image segmentation, and thus after having understood the image, one can easily spot the inconsistencies.

## 3. Inpainting Forgery Detection Mechanism

In the following sub-chapter, image forgery detection mechanisms are presented, with a deep focus on detecting only forgeries generated via the inpainting process (either manually or automatically). Regarding the overall forgery detection mechanism, there are existing reviews in this area, and a few that are relevant and have gone into a lot of detail are as follows: [3,5,51,52,53]. Their focus is on generical forgery determination methods, while in the following paragraphs, the focus will be closer to the proposed methods for inpainting determination. The following chapter will be organized as follows: state-of-the-art inpainting detection shall be presented—first with classic methods focused on specific artifacts and later with deep learning methods.

We will start with a small review of diffusion-based inpainting detection. Due to the nature of these methods, which usually fill a small area, there have been very few attempts to try to identify images that have gone through diffusion-based inpainting. In [54], the authors realized that in diffusion-based inpainting, the Laplacian acts differently in untouched vs. touched regions. Based on this observation, they proposed using two discriminating intra- and inter-channel variances. Of course, they needed to add some extra post-processing to eliminate false positives, especially on images that had under- or over-exposure. In the context of machine learning, there are only two attempts that were identified [55,56]. One of the papers presents a feature pyramid network for the detection of diffusion-based inpainting, while the other suggests using the same observations that Laplacians are usually affected by inpainting and thus a feature extractor and ensemble classifier should be able to detect those regions. Another recent research [57] suggests using a local binary pattern mechanism to detect only diffusion-based inpainting.

Before going deep into inpainting forgery detection, let us revise the copy-move forgery detection mechanisms. In a 2012 article [58], the authors presented a framework for determining copy-move blocks from images (CMFD)—see Figure 21. Their focus was on reliability and exploring various mechanisms for determining duplicate areas. Their proposed algorithm consists of the following main steps:Feature extraction (either via block-based or using variants of key points detection like SURF/SIFT). From their analysis, the Zernike moments feature extraction gives the best overall results. Also, the algorithm is less influenced when the copied area is either shrunk and/or rotated. Additionally, the algorithm (feature extraction mechanism) seems to work on various attacking vectors like resizing, jpeg compression, blurring, etc.Matching—here, they suggested a variety of methods including kNN, brute force, etc. Based on the analysis of the authors in [58], kNN gave the best resultsFiltering was undertaken to ensure the “estimated” blocks do not violate some other constraints (like the distance between them, etc.)

Analyzing the above-proposed framework, it can be observed that it works well on the copy-move scenario because it assumes that the copied area is at least bigger than the copy-paste algorithm’s patch size used in detection. However, due to their nature, inpainting algorithms do not copy a fixed-size block, but rather, at each step they try to decide what is the best patch to fill in the missing information. The following will introduce several elements that form the above framework, and other variants that were added to it, which are unusable for these types of forgeries. Summarizing the various inpainting techniques, the following was observed:Diffusion-based/PDE/variational-based—they are not suitable for filling large areas, and usually they do not copy patches but rather propagate smaller information into the area to be reconstructed. Thus they do not copy patches but rather fill the area with information diffused from the known region. So, applying a block-based detection will yield no results, as there are no copy regions but rather diffused areas. Still, some traces can be analyzed, but they are more inconsistencies in blurring, CFA, and other camera/lens properties.Patch-based—at first, they seem well suited to the above-mentioned framework. They work well if the forged region contains a lot of texture but fail in case the object removed is rather large or surrounded by a continuous area (like sky, water, grass, etc.). But upon a closer look, this method may give unreliable results due to the inpainting procedures: usually, patch-based methods reconsider the filling area at each step, and the currently selected patch may vary in location from the previously selected patch. Thus, for the forgery method, if it selects a larger area that contains several inpainting patches, it will not be able to properly determine a similar area. On the other hand, for the forgery method, if a smaller patch is selected, two aspects might arise—one will be the speed of the method and the other will be the necessity to add some other mechanism to remove false positives.

### 3.1. Classic Image Forensic Techniques

Considering the above constraints, various authors have proposed some ways to circumvent these problems. One of the first proposed methods to detect object removal was [59]—the detection of exemplar-based inpainting (DEBI). The authors proposed a zero-connectivity algorithm and added fuzzy logic to exclude areas likely to be false positives. They make the following observations from the Criminisi paper: because of the way the object removal algorithm works, it will end up introducing some patches that are going to be disconnected. Their algorithm proposes scanning the entire image (or rather a region of interest) and comparing patches and if they are found similar, passing them through a fuzzy logic semi-trapezoidal membership function. When comparing patches to determine if they are similar, they proposed the following:Compute all patches from the image (or from ROI). They are called suspicious patches.For each patch in the suspicious patches, apply the following algorithm:○Compute all the other image patches, and compare each one of them to the suspicious path.○Create a difference between the two patches.○Binarize the difference matrix○Find the longest connectivity (either four-way or eight-way) inside the binarized matrix.○Compare the obtained value with the maximum longest connectivity obtained for the suspicious patch.In the end, apply fuzzy logic to exclude some of the false positive cases.

The algorithm works well for some test scenarios, but it has some serious drawbacks, especially if the targeted image is not altered via the Criminisi inpainting algorithm but rather with a newer variant of patch-based (or ML-based). First, the computation effort is very high. If the ROI is not specified (and usually this is the case), then for a given image of size *M* × *N* and a block size *B*, the algorithm must compute (*M* – *B* + 1) * (*N* – *B* + 1) patches. Then, it must take each patch and compare it to the other patches, and thus the time taken grows exponentially. Also, fuzzy logic works well if the ROI area is given, but in case the entire image is ROI, then it fails. Also, the algorithm provides a lot of false positives for larger areas.

Several years later, in [60], they enhanced the original paper from 2008 with some extra steps. The first major change the authors made was to enhance the search mechanism. They proposed a two-step search. First, they used a weight-transformation of the blocks. The main idea was to be able to increase the performance of the block comparison by first grouping similar blocks into a structure like a dictionary, where several blocks that resemble each other are categorized in the same “key” of the dictionary. Later, a comparison of blocks was made only per key, which meant they took all blocks that belong to the same key and tried to see what the best match is. The authors compared several mechanisms for various methods of block grouping like equal numbers, even numbers, odd numbers, prime numbers, and a weight transformation proposed by them.

Another relevant improvement in this search mechanism was later proposed by authors in [61], where they only used the central pixel as the key, rather than computing a value by applying the weight transformation. After that, they used the information to search for similar blocks inside the grouped blocks. They again used the zero connectivity algorithm proposed in [59] for block comparison. After this operation, they proposed a vector filtering mechanism to eliminate a lot of false positives. In short, they imply that if two blocks are similar, their distance should be as short as possible in a normal situation. If the distance is greater than a threshold (the authors in all papers do not present the values used), they mark that block as potentially forged. The next step in their research is to decrease even further the false positive rates. For this, they noticed that the Criminisi algorithm copies patches from different areas. Based on this observation, they proposed using a multi-region correlation. In short, blocks identified in previous steps are grouped (if they are connected), thus obtaining some bigger regions. For each region, the reference blocks are identified. If there is a self-relationship group or a pair-relation group, then this area is marked as not forged. Summarizing this last step, the region created by blocks that are connected is marked as forged if there are referenced blocks from other regions (the authors proposed a value of 3—thus if the region references at least three other regions, then that blocked is marked as forged). Again, as a result, the method produces overall good results for Criminisi object removal cases, but it still has problems with diffusion-based inpainting or with enhanced example-based object removal methods. Two other potential improvements can be addressed in terms of speed—it still takes a lot of time to calculate the hash value of each block—and the fact that they apply zero connectivity mechanisms, which is good for areas in the inpainted region but applying some mechanisms somewhere in the middle of the inpainted region might seem like a redundant step. Last, but not least, based on all the analyzed pictures, it produces good results when the inpainted area (removed object) is either in a fully textured region or the object removed is at the boundary between two texturized regions.

Another improvement was suggested in [62], in which the authors suggested that a jump patch approach increases the overall results. Later in [61], the authors suggested several improvements to the original 2013 paper [60]. The first difference compared to the original paper is that they improved the two searches by changing the weight approach with a simpler and faster approach. They took “only the central pixel value of each block as the “key”. Secondly, after computing the differences matrix, they suggest replacing the AND with an OR between the color components, and in this way, the recent paper makes a stricter comparison of blocks. This algorithm seems an important improvement compared to the previous ones, but it still suffers in different areas:They have only evaluated a recent Criminisi variation paper and not the state-of-the-art (at that time) methods for inpainting (and especially for object removal), as they have used [63].The computation effort is still very high. Again, applying the GZL in the middle of the forged area seems a little too exhaustive and will not affect the overall results.

The same authors, several years later, proposed to extend the above framework with a machine learning-based approach in [64]. If the above frameworks do not generate any false positives, images are fed into an ensemble classifier. For feature extractions, the authors relied on the fact that there must be a generalized Gaussian distribution between the DCT coefficients of various blocks. Other authors tried similar approaches to what the authors proposed in the CMFD framework. E.g., [65] suggested using Gabor magnitude as a feature extraction to extract the features. The rest of the methods resemble the proposed CMFD framework (like block comparing, sorting, and detection). The interesting thing regarding this paper is the claims that the method is robust against inpainting methods, although they did not mention what method they applied for object removal. Another interesting method was proposed in [66]. Here, the authors analyzed the impact of sparsity-based inpainting on color correlations. They noticed that a modified canonical correlation analysis might be able to properly detect these artifacts. An interesting direction was in analyzing the reflectance of the forged and non-forged areas. The authors in [67] identified that some inpainting methods leave these traces, and they suggest that this approach should be combined with the CMFD framework. Below in Table 1. Classic inpainting forgery detection, a summary of the methods described earlier is depicted.

### 3.2. Machine Learning-Based Methods

Before we start a deep review of deep learning methods for image inpainting, we would like to summarize all the classic methods that constitute the basis for the deep learning approach and their limitations. The first focus is on analyzing the physical traces. Here, some of the base ideas are searching for blur inconsistencies, chromatic aberrations inconsistencies, or radial lens distortions (for a more in-depth review we strongly suggest the works of [5,37]). Usually, the methods suggest splitting the image into patches and searching for inconsistencies in one of the three areas mentioned earlier. The work does not target a specific type of forgery because it relies on the assumption that whatever type of forgery method was applied (copy-paste, resampling, inpainting, copy-move), they all disturb the overall balance of the above properties. Mentioned here are a few relevant papers that are the basis for machine learning methods:For blur inconsistencies, one of the most cited papers is [68]. They rely on the assumption that if the targeted original image contains some blur, combining parts from other images will make the blur inconsistent. They propose a method to analyze the gradients and detect inconsistencies among them. Of course, the method does not give good results in case the target digital image does not contain some blurred artifacts.Some other researchers focused on other camera properties like lens traces. The authors in [69] postulated that in some cases, it is possible to detect copy-move forgeries, in particular, by analyzing the lens discrepancies at the block level. Their method detects edges and extracts distorted lines and uses this in a classic block-based approach to analyze discrepancies within the image. The problem with this approach is that if the targeted area is either too big or too small, the results yielded are not very satisfactory. There is also another problem with low resolution images because they tend to yield false positive results.A very good camera-specific parameter that was heavily studied is the noise generated at image acquisition. Several authors have proposed different mechanisms to detect inconsistencies of block noise levels. Some authors even went in the direction that suggests that based on noise patterns, they will be able to uniquely identify camera models. To name a few, some of the most cited works are [70,71,72,73,74,75,76,77]. For e.g., in [70], the authors suggested computing noise for non-overlapping blocks and then unifying regions that have similar noise—thus partitioning the image into areas with the same noise intensities. The authors suggested using a wavelet and median filter approach on grayscale images to compute noise. Of course, the main limitations of these methods vary from false positives to the impossibility for these methods to detect if noise level degradation is very small (a lot of anti-forgery mechanisms can exploit this method).Color filter array methods or de-mosaicking methods (CFA) rely on the observation that most cameras capture only one color per pixel and then use some algorithms to interpolate these values. The forgery detection mechanism based on the CFA detects inconsistencies at block levels between the patterns generated by the CFA. One of the most cited works is [71], in which the authors propose using a small block (up 2 × 2) to detect inconsistencies in the CFA pattern. They extract the green channel from the image, calculate a prediction error, and analyze the variance of the errors to mark the non-overlapping block as forged or not. The method yields good results as the original image does not suffer from some post-processing operations like color normalization.

One of the first machine learning-based methods is the work of Linchuan Shen, Gaobo Yang, Leida Li, and Xingming Sun [72]. Here, they propose a support vector machine classifier composed of the following features: local binary pattern features, gray-level co-occurrence matrix features, and gradient features (they suggest using 14 features extracted from patches). One key aspect of their research is the robustness of this method, especially on post-processing operations (like jpeg compression, scaling, noise, etc.). Other variants of ML-based detection are [73], in which they employ a standard CNN model to be able to detect tampered regions [74], that suggests using four Resnet modules to extract features from image blocks, and a later one that employs a hybrid model of LTSM and CNN [75]. Like the above, the work in [76] also uses a simple CNN and extracts general features from inpainted vs. non-inpainted areas. The main problem with the above methods is that they either evaluate the network in the old Criminisi paper, or they randomly select a center area in real images and apply the latest image inpainting methods. Neither have the potential to be used in real-life scenarios. 

In the last years, the focus regarding detecting inpainting images is on applying higher and more complex machine learning models with some strong feature extraction mechanisms. For example, in [77,78] the authors suggested that the noise variance disturbs the inpainted area, and thus by applying three Resnet feature blocks in a multi-scale network, they obtained very good results. They are used for assessing the latest state-of-the-art deep inpainting methods, but they use random masks for applying inpainting methods and are thus not very realistic approaches. Another fusion-based approach was proposed in [79]. Their work was actually inspired by [74,80,81], and they suggest using three enhancements blocks: a steganalysis rich model to enhance noise inconsistencies, pre-filtering to enhance discrepancies in high-frequency components, and a Bayar filter to enable the adaptive acquisition of low-level prediction residual features. After these blocks, the authors used a search block to detect the areas based on the three enhancement blocks. The last step is a decision block because there is usually an inconsistency in how pixels are classified. Taking a closer look at the results, the authors obtained very good results compared to previous algorithms. An interesting approach was also used in how they generated data—they incorporated various datasets of pictures and used several inpainting methods to generate forged data. The problem with their approach is still that the mask data used for the inpainting process is generated randomly and not in a realistic manner. In the same category as the IID-NET methods, others have suggested [82] incorporating more enhancement blocks to make the detection more reliable, and for a more general range of forgeries, they used several datasets some of which present not very realistic tampering (more on the datasets in the next chapter).

Another more complex architecture was added in [83]. The authors based their network architecture on the work of [84]. Again, like in previous machine learning approaches, the authors try to create a method to suit all types of forgery like splicing, copy-move, and inpainting. The novelty in [83] was adding a top-down and a bottom-up path encompassed by another color correlation module called the spatio-channel correlation module. In the top-down area, they tried to detect features at different scales, while in the bottom-up part, they started from the mask to find and strengthen the correlation between forged and non-forged areas. They claimed to obtain top SOT compared with other systems like [85,86,87,88,89,90,91,92,93,94]. Again, the authors opted to create their own dataset to tackle various forgery types like splicing, copy-move, and object removal. An interesting thing was that they did not compare their work with the other that claimed to be SOTA [79]. Also, the work based on noise inconsistencies [77] appears to surpass these approaches in terms of results but not in terms of speed and model complexity (the authors in [83] claimed to use a small network model with approximately 3 M parameters only).

Summarizing the above-presented machine learning, two trends regarding detection types can been seen: algorithms that focus solely on detecting inpainting (object removal) forgeries; and methods that try to tackle all kinds of forgeries. From the dataset point of view, most of the research uses either manually generated datasets by applying random masks or old datasets. From the results point of view, there are several areas that offer good results, and the most promising ones are in the detection of mismatches in either the noise and/or color information presented. In Table 2. Machine learning-based inpainting forgery detection, a summary of the presented data above is shown.

## 4. Image Inpainting Datasets 

Based on the previous deep analyses, it was observed that most of the proposed methods employ a self-made dataset. This requires lot of effort when trying to compare with previous SOTA to check if the newly proposed methods indeed add improvements. Classic inpainting detection mostly uses the initial work of Criminisi to be able to detect (with some small exceptions). On the other hand, deep learning methods have tried to use newer variants for inpainting/object removal. The problem with this approach is that each machine learning forgery method employs another inpainting mechanism they consider relevant. Of course, there is another aspect, especially with deep learning methods: usually, authors try to compare their work with the latest inpainting methods, and thus there is also the need to have an open dataset that can grow bigger and bigger. Also, another aspect is related to automatic vs. human-based generated datasets. For e.g., there are very few research articles that take into consideration manually well-crafted forgery using various tools (like Photoshop). To circumvent these limitations, several authors tried to standardize their datasets. Our focus will be to present datasets that pertain to inpainting/object removal detections (but if there are some datasets that have several other categories, they will be presented as well). For a deep review of generic datasets used in forgery detection systems, the works of [5,86] are recommended. To make the distinction clearer, the presented work is focused only on one object removal/inpainting manipulation. In the following sub-chapter, the current object removal/inpainting portion from generic datasets is analyzed, while in the second sub-chapter, the focus is on the dataset created specifically for the task of detecting these types of manipulation.

### 4.1. General Forgery Datasets

One of the first truly realistic forgery datasets is in [58,87]. The MICC dataset was one of the widely accepted datasets for image forgery. It consists of four sub-datasets called F220, F2000, F8 multi, and F600. The CMFD dataset contains updates that were undertaken manually using professional tools. It is a generic dataset for copy-move scenarios and not only object removal use cases. Another relevant property worth mentioning is the fact that the dataset is grouped by the type of camera that acquired the pictures. Upon analyzing the literature for usage of this dataset, it was observed that the dataset was not referenced, probably because it contains large images and using a block-based comparison needed a lot of time to compute. At the same time, authors in [88,89] proposed a similar dataset focused on copy-move. Still, some of the pictures can be considered as having object removal rather than copy-pasting and thus constitute a good measure to test inpainting methods. What differentiates these datasets from the CMFD one is the fact that here, the authors proposed some variants of the forged ones that undergo some post-processing anti-forensic measures like jpeg compression, blurring, and noise adding. An important factor to mention is that the images in the Comofod dataset were generated using Adobe Photoshop (especially for the splicing artifacts). The interesting, unique factor for the Comofod dataset is that the authors tried to have images that naturally contain similar but genuine objects. For the Casia dataset, the authors introduced two sub-datasets—namely Casia V1 and Casia V2. The second version contains pictures with several sizes (up to 900 × 600), and it is important to note that in some instances, they add some pre-processing and post-processing operations like resizing, distortion, blurring, noising, etc.—thus making detection harder. Some of the generated images are quite realistic—the copy region is not a simple copy-move region, but the copied region is also resized and rotated and goes through additional processing. It is also important to notice that the original dataset did not offer masks, so the following [90] can be used as a reference for masks and some filtering of the initial dataset (mis-categorizations or duplicates removal). In the same category, newer versions are the COVERAGE dataset [91] or the work of Pawel Korus in his papers [92,93]. The COVERAGE dataset, although small (100 images with an average size of around 400 × 500 pixels), tries to make things even harder by suggesting pictures that contain similar but genuinely valid objects. The “copied” items are inside the images, thus they already resemble multiple instances of similar objects, and the difficult task is in detecting real and faked areas vs. similar instances. The work of Pawel Korus contains one of the most realistic forgeries, which focuses on several types of different cameras (some images were his own, and some were taken from [94]). The sizes of the images are bigger than those in previous work (1920 × 1080), but in total, there are only around 220 forged images, and the forgery methods vary from copy-move to object removal. It is important to mention that they used GIMP and Affinity for generating the forgeries, and the dataset consists of four different camera models. The authors suggest that when copy-move or object removal is applied on an input image, it distorts the overall noise between patches inside the image. Another relevant dataset that was started in 2015 with the Media Forensic Challenge for Darpa [95,96] now consists of millions of media that cover several forged mechanisms including both images and videos. Most media data is collected from the internet from various sources and in different contexts to ensure a more generic dataset. From these millions of media, approximately 200k images are what the authors called high provenance—the authors ensured that the images have not undergone any forgery attacks. For this large dataset, the authors employed experienced people to perform various forgery attacks (and they were not limited to using only specific tools for forgery generation). The MFC dataset consists of several sub-tasks like image/video manipulation localization, splice image/video manipulation, provenance, graph building (detecting all the steps and sources a target image has undergone), GAN, etc. At this point, it is still considered one of the most referenced forgery datasets both from quality and quantitively perspectives. The MFC dataset (2019 version) has approximately 180, 000 original images, obtained directly from cameras, thus ensuring that the images are not manipulated in any way and all device specific settings, like, e.g., ISO, are known beforehand. Also, to ensure a large diversity of the images, the authors used more than 500 different cameras to have a large pool set of different noises that can be used for training (PRNU-based algorithms). From this, approximately 16k images have been altered. Still, the main problem is that a small part of this dataset is focused on object-removal (content aware fill)/inpainting tasks. Even the authors in MFC recognized that they have added several methods under the generic umbrella of copy-move alterations. Table 3 (General forgery datasets for which a subset can be used for inpainting/object removal) displays a concise overview of the facts presented above.

### 4.2. Image Inpainting Specific Datasets

One of the first datasets with a specific division of object removal was that in [97]. The Defacto dataset consists of the three main forgery use cases: copy-move, object removal, and splicing. For each type of forgery, they have a subset of images. They indeed have an exceptionally large dataset of about 25,000 images, which were generated using [98]. To generate the dataset, they actually used the MSCOCO [99], which contains descriptive information. Next, they randomly chose a descriptive region from each image and expanded the raw mask. They then performed the inpainting technique and discarded photos that had standard deviations below a specified threshold for the inpainted regions. The issue with the dataset lies in the random generation of masks, resulting in blurred and immediately detectable forged images in places with high texture or inaccurate MSCOCO descriptions as well as at border regions. This detection may be performed even through human observation. Although the dataset has gained traction with at least 34 methods utilizing it since its inception, all usage is primarily focused on either splicing detection or copy-move detection.

Using the same methodology as the previous authors is the dataset entitled IMD2020 [100]. In this dataset, the authors took 35k pictures from Flickr, manually reviewed them, and then used a combination of classic inpainting from OpenCV and a machine learning-based model for inpainting [101]. One issue pertaining to this dataset is the incomplete differentiation between the original and forged areas in the masks. If the pixels are compared from the original vs. forged areas, there are a lot of differences not only in the altered region (the images underwent some post-processing, which was not mentioned in the paper, or the images were altered as part of the inpainting procedure). Another problem with this dataset is that not all forged regions are realistic.

One of the most recent datasets proposed especially for inpainting (object removal) detection is in [85]. They presented about 10k images, but as a novelty, they did not target one specific inpainting method but rather proposed 10 methods (6 classic method and 4 machine learning-based). Still, the same problem with the masks generated for the inpainting process was observed. Because they have generated masks with varied sizes and shapes and did not consider the nature of each image, some of the generated inpainting was not very satisfactory. Also, another problem is that the dataset consists of small images—all have the same 256 × 256 size. 

Table 4 (Image inpainting forensic dataset) provides a succinct summary of the data.

The difficulties identified with the present datasets can be summarized as follows:Each approach for detecting object removal (inpainting) is accompanied by its own customized dataset. Authors employ this methodology to evaluate their detection systems on distinct customized datasets.Datasets are created using segmentation masks but due to the automatic image selection process, not all cases are pertinent. For instance, while the mask is accurately chosen, the current inability to remove the object is owing to limited context and a heavily texturized area. Put simply, when certain limitations are present, there are no available techniques to populate the area with pertinent information.Another aspect is the way items are removed through the process of inpainting. Authors employ a variety of methodologies for removing, including both older and modern inpainting techniques. However, there is a lack of a systematic and backward-compatible approach for testing varied inpainting methods. As will be shown later in this paper, depending on the context, certain older inpainting approaches may be more difficult to detect than newer ones and vice versa.

## 5. Results and Discussion

The subsequent section entails an examination of the outcomes derived from the diverse inpainting detection techniques employed on different image inpainting processes. The primary emphasis is on the datasets. While there are other forgery datasets available, such as Casia and MFC, they lack specificity as they do not solely focus on image inpainting or object removal. Furthermore, based on the previous analysis (refer to the Dataset chapter), it was noted that each forgery detection method often has its own specific dataset for training/testing. To address this, Google’s Open Images Dataset V7, released in October 2022 [102], was used. Manually, 400 images were selected with additional segmented masks—only one mask (object) was used per image. The selected segmented masks were chosen to not be in very texturized areas—this limitation was imposed because almost all inpainting methods have problems filling highly texturized areas. Additionally, because forgery methods do not work properly on big images, the selected images should be a maximum of 1024 × 1024 in size. Another relevant aspect is the chosen segmented object. The dataset tries to cover the distinct sizes of objects to be removed from small ones to large objects present in images. The last relevant aspect is that regarding the provided masks from Google’s dataset, it was observed that some of the masks were close to the defined borders, so to enhance the inpainting methods results, a dilation was added with a kernel of 5 × 5.

To accurately assess the performance of several detection methods, a dataset was generated based on various inpainting methods. This work is inspired by [79], in which the authors selected several inpainting methods and proposed a forgery mechanism to learn the intricacies of all those inpainting methods. The main distinction lies in the absence of a randomly generated mask in the present study. Instead, an already segmented object was extracted from the image. Additionally, the same inpainting method was applied to multiple inpainting methods, allowing for a comprehensive evaluation based on the image-mask pairs. By examining the various inpainted outputs, the effectiveness of each forgery method in detecting traces can be thoroughly assessed.

For the inpainting methods, five different inpainting methods were used: the Criminisi original method, an improved version of the patch match algorithm [103]; professional editing tools–GIMP [104]; and two machine learning-based methods—Lama [40] and a newer improved version called MAT (mask-aware transformer for large hole image inpainting) [105]. The selection was made to ensure that representative methods are used in relation to editing tools, most relevant and cited patch-based methods, and state-of-the-art methods in terms of inpainting using neural networks.

For forgery detection, six different methods were selected to evaluate them using the above dataset. The first pick was the CMFD method proposed in [58]. Although the CMFD is focused on generic copy-moves, we wanted to check how well the method is able to detect inpainted areas. As parameters, Zernike moments were used with a block size of 13 based on the author’s suggestions. A classic object removal detection method was selected as well [59]. The choice of this older method with a newer variant was dictated by the that fact that a newer variant like that in [61] improves both speed and accuracy by eliminating a lot of false positives. The next four methods selected were machine learning-based. The first on the list id the Mantranet method [106] as it is the basis for numerous newer methods. Based on the authors’ claims, the method should also detect object removal because they have trained it with images generated by OpenCV inpainting methods and additionally used more than 300 different classes—like blur inconsistencies, Gaussian inconsistencies, etc.—to train the network. The next method based on the Mantranet is the IID network [79], which solely focus on inpainting detection. Additionally, two more newer generic forgery methods were included: focal [107], which adds a clustering method to be able to differentiate between forged and not forged areas in the image; and PSCC-Net, which encompasses a spatio-channel correlation module to be able to focus on various traces. All the neural networks-based detection tests were generated on machines with a NVIDIA Quadro RTX 8000 video card.

In Figure 22, the original image and object to be removed are presented. As noticed, in Figure 23, the first result based on [19] does contain some visible artifacts, while the others are able to complete the image in a very natural way. Based on [104], by duplicating a section with a lower luminosity and afterwards contrasting it with the overall context of the image, it becomes possible to ascertain that the said area has been replicated from a neighboring location, so a simple block lookup comparison determines the similar regions. In [103], the regions nearby were duplicated and interpolated more smoothly, but it is noticeable that it introduced a blurring artifact on the region of the removed object in the highlighted area in Figure 24.

Next, an analysis is performed on each of the six algorithms: DEBI [59], CMFD [58], IID [79], Mantranet [106], PSCCNET [83], and focal [107]. In Figure 25, the visual results are presented.

Because DEBI [59] uses the idea of block comparison, it can easily be observed that the method is able to detect some clues on images produced by Criminisi [19], Gimp [104] and Non-local patch [103]. The results indicate that some regions are tampered because all three inpainting methods work rather similarly by copying various patches from different regions. An interesting observation is that except for the Criminisi method, all methods affect the overall pixel intensity not just in the target masked area. This is why the method by [59] can detect different regions—even some that are false positives as one can notice in the above figure in the c picture. Somehow, the results are expected for the machine learning inpainting methods. Because machine learning methods do not copy a patch but rather try to synthetize, the block-based approach is not able to detect any similar blocks. A solution we analyzed was to search by similar blocks within a given delta for pixel differences, but by doing so, a lot of false positive results were received. Comparable results were obtained also from the CMFD framework because the detection method is identical, and they work by comparing blocks and applying some filtering logic on similar blocks.

The next analyzed method was Mantranet [106]. As it can be noticed in Figure 26, the detection method works with reliable results using classic inpainting methods but poorly with machine learning-based inpainting methods. Also, of relevant note, especially in the computation of F1 score, precision, recall etc., is the fact that [79,83,106] provided results of images with gray-level intensities. This means that if the region is perfectly white, there is a high confidence that the pixels are tampered with, while lower pixel values give lower confidence. In the evaluation metric measurements, an analysis was performed with different values of these pixels intensities to see how they affect the overall results.

Using IID [79], the results for this image with all five different inpainted images were very promising, with a small observation that for [103], it only detected the surrounding area as it can be seen in Figure 27.

Using PSCCNET [83], the results were poor, but on the other hand, interesting results were observed using the focal [107] method. The results for the focal method are presented in Figure 28. The method can successfully detect forged areas for block-based inpainting methods. It behaves strangely with the machine learning-based method, where it detected artifacts incorrectly related to the object, which was not removed but altered by the inpainting method.

In the context of detecting forged images, especially when dealing with subtle manipulations like inpainting, it is crucial to use metrics that accurately reflect the effectiveness of the detection method. F1 score, precision, recall, and intersection over union (IoU) are commonly used metrics in this field. Let us define each metric and explain why they are important:Precision is the ratio of correctly identified positive cases to all cases identified as positive. Precision measures the accuracy of the detection in identifying forged images. High precision indicates that most of the images identified as forged are indeed forged, which is crucial to avoid mislabeling authentic images as fake.Recall, also known as sensitivity, is the ratio of correctly identified positive cases to all actual positive cases. Recall measures the ability of the detection method to identify all forged images. High recall is important to ensure that most, if not all, forgeries are detected.The F1 score is the harmonic mean of precision and recall. The F1 Score provides a balance between precision and recall. It is particularly useful in scenarios where an equal importance is given to both false positives and false negatives, which is often the case in image forgery detection.Intersection over union (IoU) is a measure of the overlap between two areas. For image detection tasks, it is the area of overlap between the predicted bounding box and the ground truth bounding box divided by the area of union of these two boxes. In the context of image forgery, IoU can be particularly relevant when the task involves localizing the altered part of the image. A high IoU score indicates that the detected area of forgery closely matches the actual forged area.

Using these metrics together provides a comprehensive assessment of a detection method’s effectiveness. Precision and recall offer insights into the accuracy and completeness of the detection, while the F1 score provides a single metric balancing these two aspects. IoU is especially valuable in tasks where the precise localization of forgeries is important, as it measures the accuracy of this localization. Following this, a comprehensive study has been undertaken to evaluate the results from the perspective of a measuring meter. Initially, an evaluation was conducted to assess the performance of the two block-based method detection techniques. Upon conducting an analysis of the F1 score, precision, recall, and intersection over union (IoU), it becomes evident that the approaches employed yield unsatisfactory outcomes. The CMFD method yielded superior outcomes in comparison to the DEBI method. It is likely that the implementation of enhancements discussed in reference [61] would enhance the overall performance of DEBI. It is important to acknowledge that the existing methodology (DEBI and CMFD) lacks the capability to detect regions that have experienced indirect replication. This is the reason why any results for either DEBI (Figure 29) or CMFD (Figure 30) are not shown in relation to machine learning-based inpainting methods. In summary, the evaluation of the Non-local patch inpainting approach using both detection methods suggests a modest level of performance based on the metrics employed. Although the current system demonstrates some accurate detections, there is considerable scope for enhancement, particularly in relation to the bounding box overlap (IoU) and the reduction of erroneous detections (precision). The poor results in terms of detecting images that underwent an inpainting process via the Criminisi method might be explained by the fact that the areas to be inpainted contained some shadow elements, and the resolution of test images is quite big compared with that of the original paper (1024 × 1024 vs. 256 × 256). Based on the measurements acquired, it is evident that the DEBI and CMFD detection algorithms, which employ the Criminisi and GIMP inpainting approaches, encounter significant challenges when attempting to detect forgeries. The methods have a notable inadequacy in forgery detection, as they demonstrate a failure to accurately identify a considerable proportion of the region (low recall). Moreover, despite their ability to identify potentially problematic areas, the effectiveness of the detection systems is often compromised, leading to a diminished intersection over union (IoU) score. The precision metric suggests that around one-third of the systems’ detections are accurate. Nevertheless, with careful examination of the subpar recall and IoU scores, it becomes apparent that improvements are necessary across all facets of the methodologies.

The method proposed in [107] demonstrates superior efficacy in detecting the inpainting produced by [103] in comparison to the rest of the methods (Figure 31). In the case of the block-based approaches, namely Criminisi and Gimp, the detection model exhibits a reasonable level of accuracy in identifying inpainted regions at approximately 60%. However, upon closer examination of the intersection over union (IoU) metric, which measures the overlap between the predicted and ground truth regions, it becomes evident that the system erroneously identifies additional sections as manipulated, resulting in an IoU of approximately 30%. The focal method applied on the non-local patch images has a commendable level of precision in its detections and effectively captures a substantial proportion of the objects that are there. The marginally reduced F1 score indicates a small imbalance, albeit without major divergence. The IoU score indicates that the model’s localization accuracy is satisfactory, while there is room for improvement. It is possible that the accuracy was affected by the wrong selection of mask for inpainting. Further information regarding the dataset talks on masks may provide insights into this matter. However, the performance indicators for the focal reveal subpar outcomes in relation to the Lama and Mat inpainting approaches and should be subject to a closer review.

The methodologies described in [79,83,106] do not produce a binary mask that may be used to identify counterfeit regions. In contrast, they present a heat map. Pixels with higher values, which are represented by white pixels, imply that the utilized approach possesses a higher degree of confidence in accurately identifying modified pixels. Moreover, as an illustration, the methodology outlined in reference [89] uses a SoftMax function on the entire output to augment the response by integrating a binary categorization of the image as either counterfeit or genuine. To adequately evaluate the three procedures, three separate approaches have been utilized to analyze the outcomes of the tests. In this study, a set of three sample values (20, 70, 127) has been chosen as the threshold for the purpose of identifying forged pixels. Pixels that surpass these values are categorized as indicative of forging, whilst those that fall below are classified as genuine. For example, when the threshold for [79] is set to 127 instead of 20, the precision increases by 3%, but the IoU reduces by around 4%. A similar trend may be observed with the Mantranet approach when the threshold is set to 127 compared to 20. With this case, there is a 12% gain in precision but a decrease of approximately 5% in IoU. In Figure 32, the results are presented with a threshold of 127.

One interesting characteristic of the IID/Matranet/PSCC-NET methods is the observation of a high recall rate accompanied by lower values of precision, F1 score, and IoU (see Figure 33 and Figure 34). This pattern suggests that the model has an excessive tendency to identify detections, successfully capturing most genuine objects. However, it also tends to incorrectly identify numerous non-objects. Furthermore, even when the model’s identifications are accurate, its ability to precisely localize objects may be compromised. This tendency may provide challenges in several situations, particularly when accuracy is of utmost importance. Additionally, it is possible that the model could acquire advantages from other optimization techniques to enhance precision while minimizing any substantial trade-offs in recall.

Based on the summary of the results presented in Figure 35, it is observed that the focal [107] performs well with the non-local-based inpainting methods, Mantranet [106] is able to detect older variants of patch-based methods, and IID performs the best with the machine learning-based inpainting methods.

The domain of picture inpainting has witnessed significant advancements in recent years. The usage of exemplar-based techniques has been a particularly intriguing feature of this evolution. Using these methodologies, it has become feasible to rectify substantial segments of impaired or absent regions within a picture. At its inception, the practice of inpainting was primarily limited to making modest modifications, such as repairing minor scratches or concealing imperfections in sensor devices. Nevertheless, at present, it has the capacity to address significantly more intricate difficulties, including the removal of considerable items. The applied strategies can be categorized into two primary groups: those that rely on partial differential equations and patch-based methods, sometimes referred to as exemplar-based methods. Contemporary photo editing software, designed for both professional and novice users, frequently incorporates sophisticated inpainting techniques. Exemplar-based inpainting can be understood as a technologically advanced and automated approach to the detection and removal of copy-move forgeries. In this procedure, segments are extracted from different regions of the image and adeptly merged to yield enhanced visual outcomes. The blending operation is a vital element of this process since it must be executed flawlessly to guarantee the cohesiveness of the finished product. In the field of convolutional neural networks, notable progress has been made in inpainting approaches, which have demonstrated superior performance compared to skilled human editors. This is particularly evident when employing content-aware filling methodologies. The capacity of CNN to construct a coherent visual storyline using limited datasets highlights its substantial potential inside this domain.

Nevertheless, the task of detecting these inpainted alterations might be a significant problem, as conventional copy-move detection methods frequently have limited effectiveness in this context. There are various explanations for this phenomenon. Firstly, the target area under consideration may be too minuscule to be effectively detected. Secondly, the modified regions may closely resemble pre-existing areas within the original image. Lastly, the inpainted areas could potentially consist of multiple distinct regions. In response to these constraints, several authors have proposed automated techniques for the detection of inpainting forgeries. These methodologies, akin to the ones employed in the detection of copy-move forgery, exploit visually identical picture patches to emphasize locations that raise suspicion. In addition, they utilize heuristic criteria to minimize the occurrence of false alarms. The heuristic rules exhibit a range of characteristics, since different authors employ various approaches, including the utilization of fuzzy logic, to address this issue. Efforts have also been made to exclude regions that lack indications of amalgamation from several geographical areas. However, the conventional approaches possess inherent constraints. The utilization of substantial computational resources and effort is frequently necessary to enhance detection accuracy by reducing patch sizes. Additionally, these methods encounter challenges in effectively mitigating false positives. In recent times, there has been an introduction of machine learning approaches to analyze disparities among various patches. An investigation was conducted to analyze noise, specifically focusing on photo response non-uniformity (PRNU), to quantify noise levels inside individual patches and detect any irregularities in the distribution of noise patterns throughout these patches. Prior research has focused on analyzing artifacts originating from the color filter array and has attempted to utilize similar methodologies employed in noise analysis. In the current epoch of deep learning, endeavors have been undertaken to devise an automated methodology capable of seamlessly amalgamating diverse artifacts and proficiently identifying anomalies. Nevertheless, the efficacy of these techniques is significantly impacted by the implementation of countermeasures such as noise reduction or addition, color correction, gamma correction, and other similar factors.

One of the key challenges in the field pertains to the limited availability of datasets, which hinders progress. Although there are several well-known datasets for detecting copy-move anomalies, the availability of datasets specifically designed for inpainting tasks, such as object removal or content-aware fill, is limited. Currently, there are just three datasets that can be categorized into this area. The restricted accessibility of datasets poses a significant obstacle in the examination and enhancement of inpainting detection techniques, impeding the prospective advancement in this captivating field of study. Hence, it is crucial that additional resources are allocated towards the creation and upkeep of extensive and superior inpainting datasets to advance the discipline.

## Figures and Tables

**Figure 1 jimaging-10-00042-f001:**
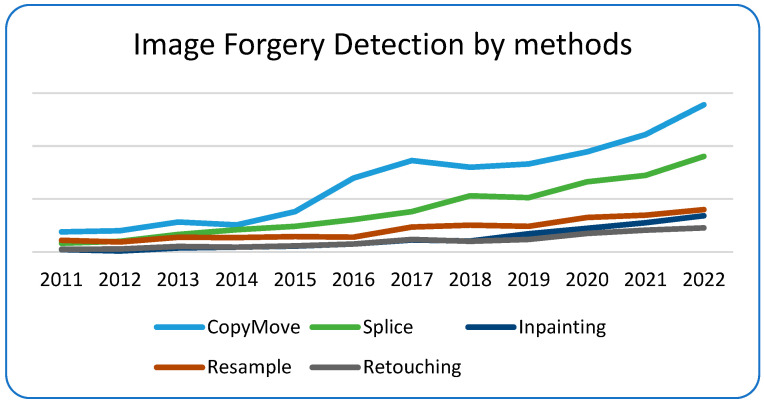
Trends in forgery detection during the last years.

**Figure 2 jimaging-10-00042-f002:**
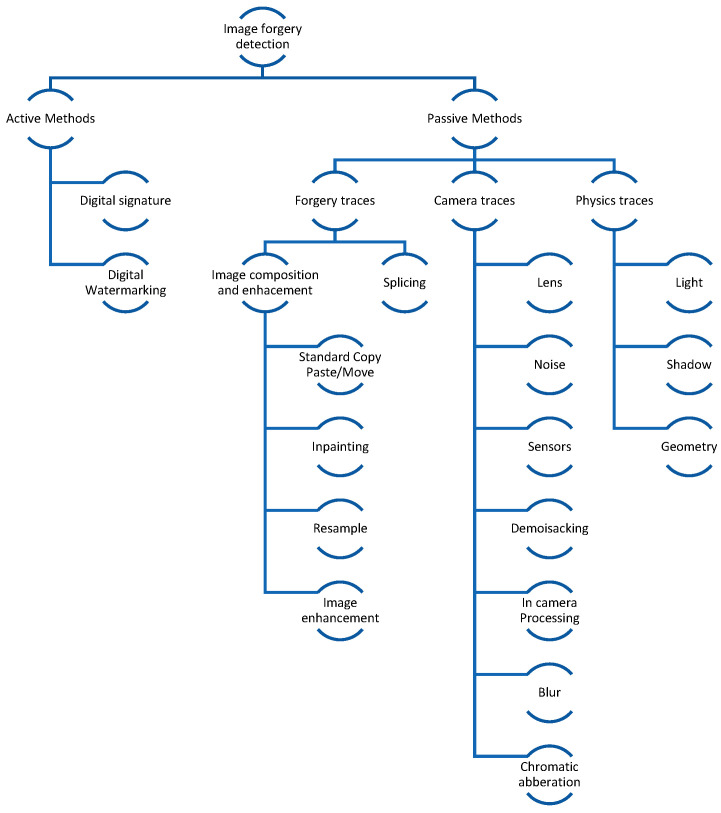
Overview of image forgery detection.

**Figure 3 jimaging-10-00042-f003:**
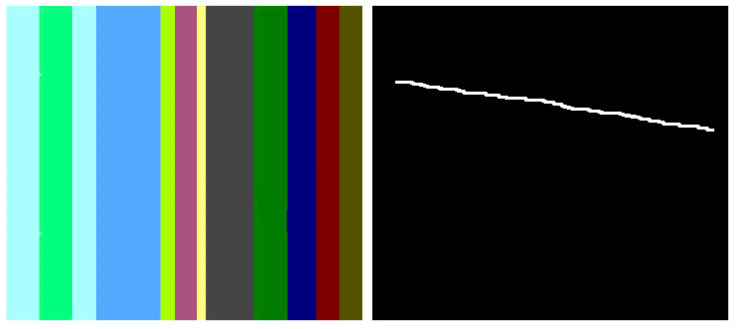
Original image and the applied mask.

**Figure 4 jimaging-10-00042-f004:**
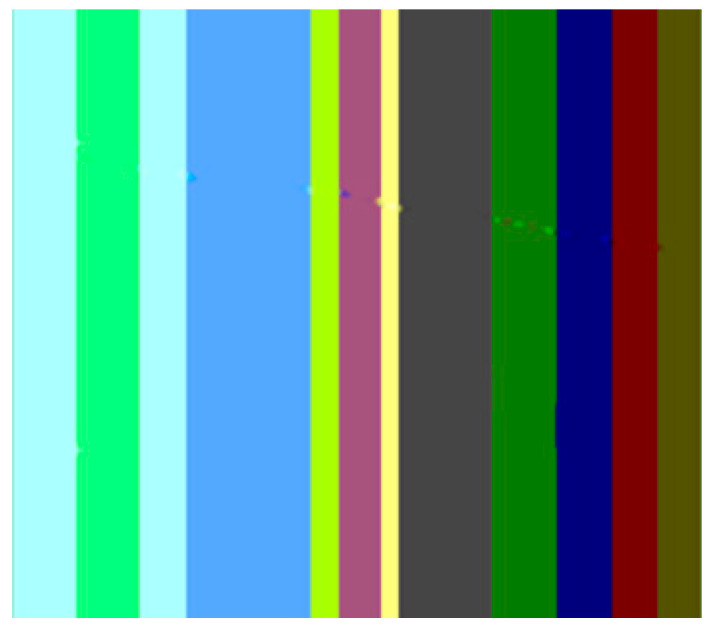
Image inpainted using Cahn–Hilliard.

**Figure 5 jimaging-10-00042-f005:**
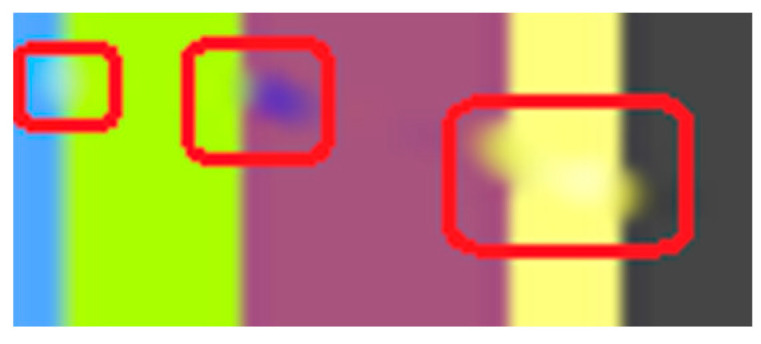
Zoomed area to emphasize discontinuities, incorrect color, blurring, and over smoothing.

**Figure 6 jimaging-10-00042-f006:**
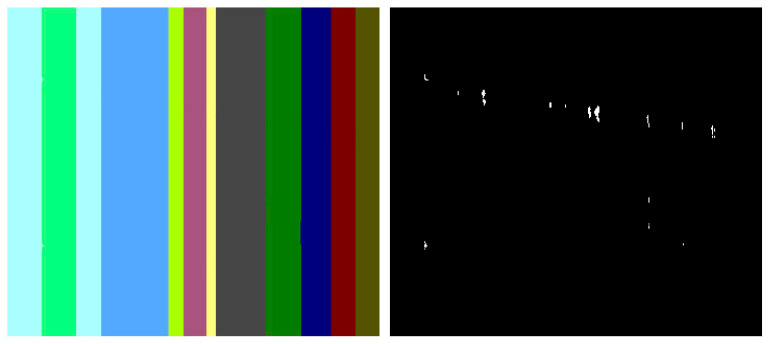
On the left side is the reconstructed image, and on the right side is the difference between reconstructed image and original image.

**Figure 7 jimaging-10-00042-f007:**
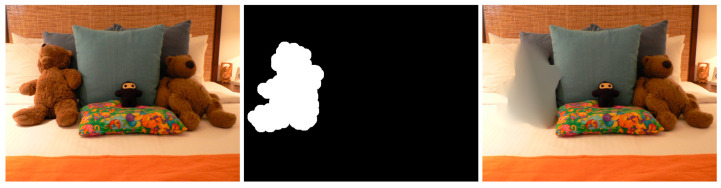
Original image, the applied mask, and the TV result.

**Figure 8 jimaging-10-00042-f008:**
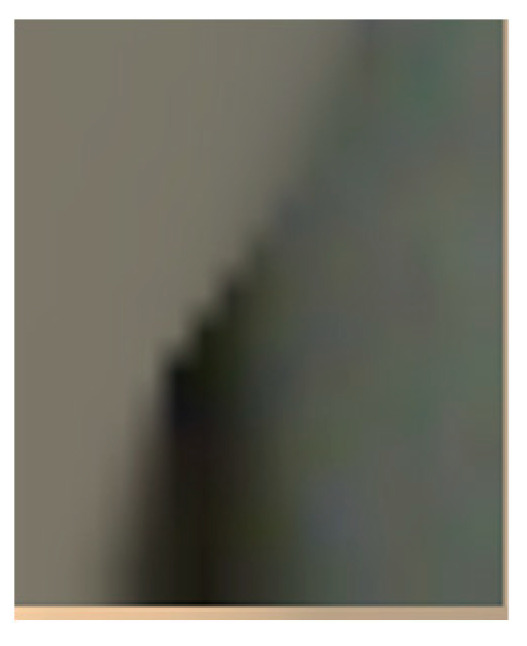
Staircase artifact seen in zoomed area from the above TV output.

**Figure 9 jimaging-10-00042-f009:**
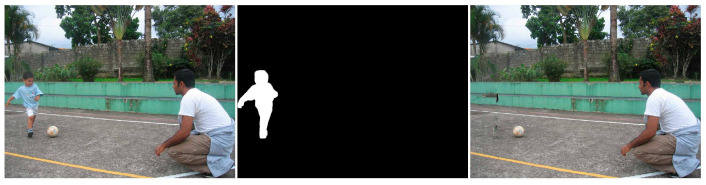
Original image, the applied mask, and the Criminisi result.

**Figure 10 jimaging-10-00042-f010:**
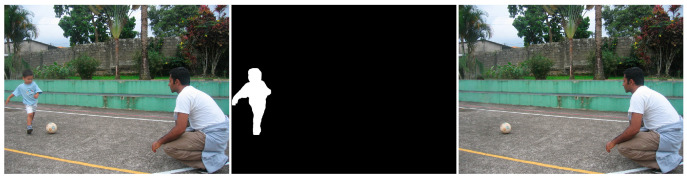
Original image, the applied mask, and the Huang result.

**Figure 11 jimaging-10-00042-f011:**
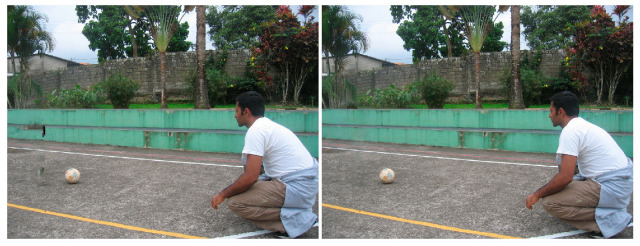
Reconstructed image by Criminisi vs. that by Huang.

**Figure 12 jimaging-10-00042-f012:**
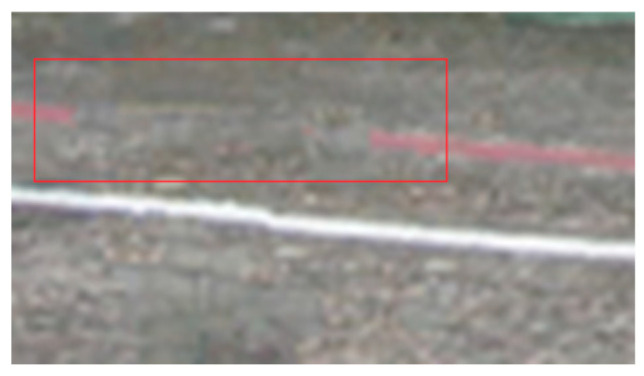
Zoomed area with incorrect color continuation.

**Figure 13 jimaging-10-00042-f013:**
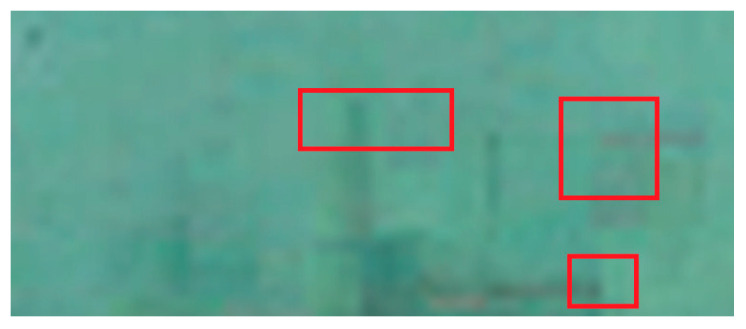
Zoomed area of the Criminisi result in which jagged areas can be seen.

**Figure 14 jimaging-10-00042-f014:**
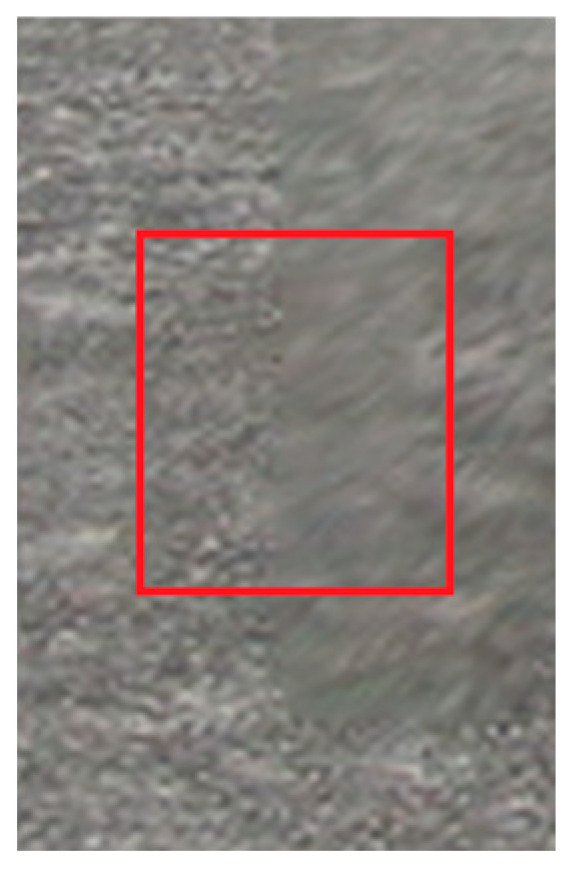
Zoomed area of the Huang result in which blurring is presented.

**Figure 15 jimaging-10-00042-f015:**
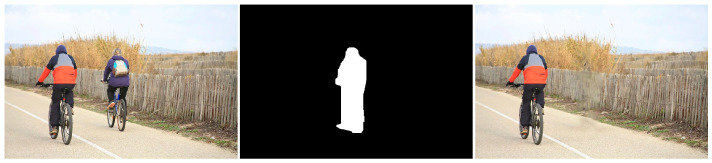
Original image, the applied mask, and Huang result on highly texturized area.

**Figure 16 jimaging-10-00042-f016:**
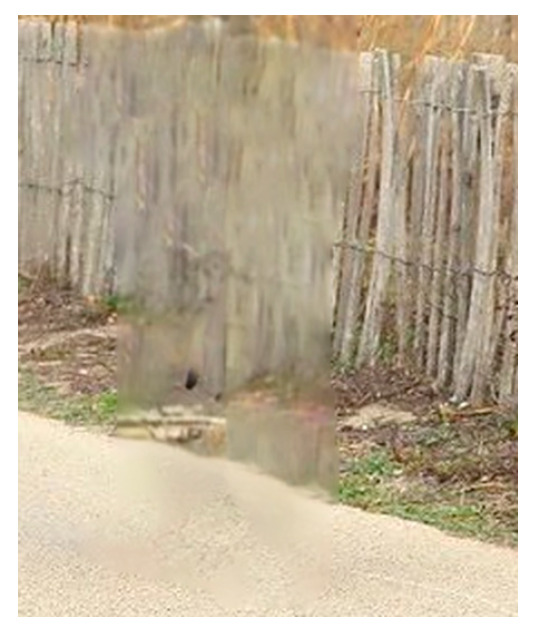
Zoomed area in which blurring artifacts are present.

**Figure 17 jimaging-10-00042-f017:**
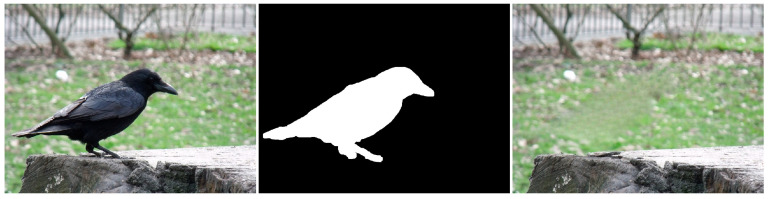
Original image, the applied mask, and Lama result with blurring and smoothing effect.

**Figure 18 jimaging-10-00042-f018:**
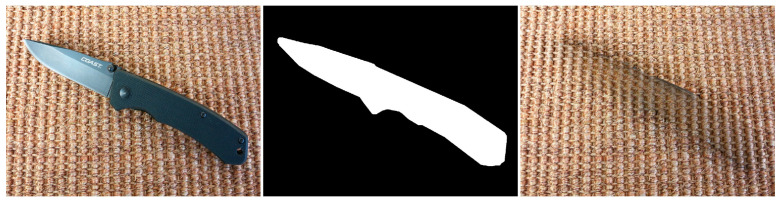
Original image, the applied mask, and Lama result with inconsistent textures and coloring.

**Figure 19 jimaging-10-00042-f019:**
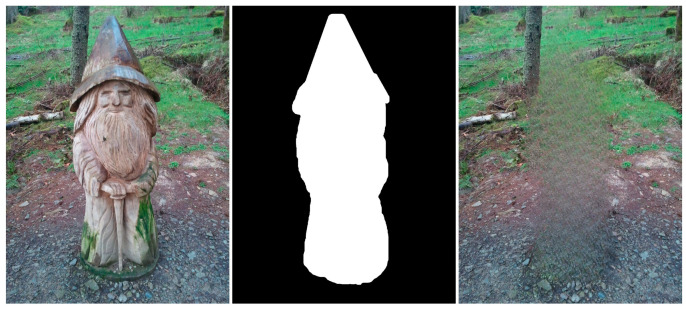
Original image, the applied mask, and Lama result with edge artifacts and structural and geometric errors.

**Figure 20 jimaging-10-00042-f020:**
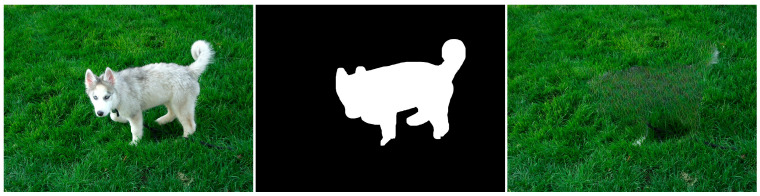
Original image, the applied mask, and Lama result with repetitive patterns.

**Figure 21 jimaging-10-00042-f021:**
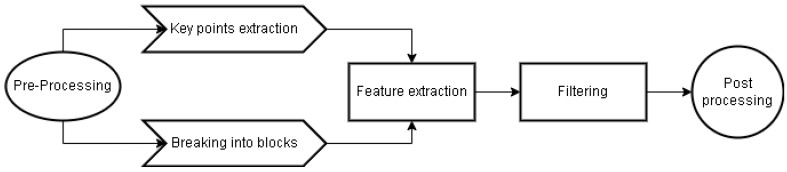
General steps for copy-paste forgery detection as presented in P. Korus paper.

**Figure 22 jimaging-10-00042-f022:**
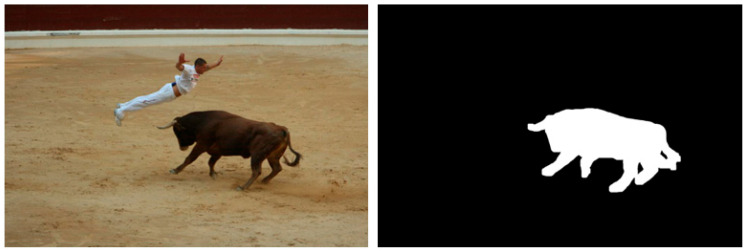
Original and mask image from dataset.

**Figure 23 jimaging-10-00042-f023:**
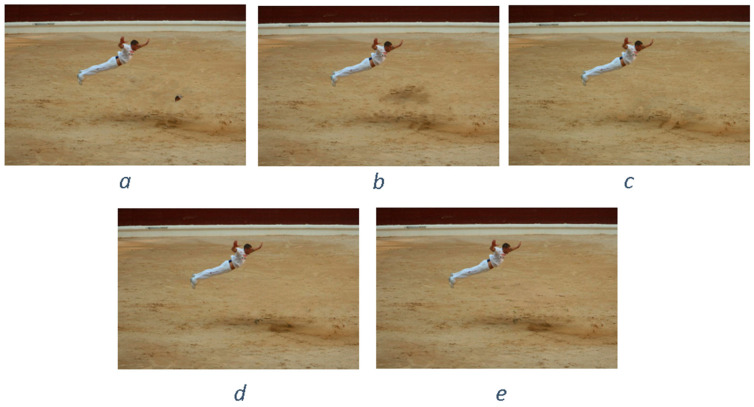
Inpainted results: (**a**) Criminisi, (**b**) Gimp, (**c**) Non-local patch, (**d**) Lama, (**e**) Mat.

**Figure 24 jimaging-10-00042-f024:**
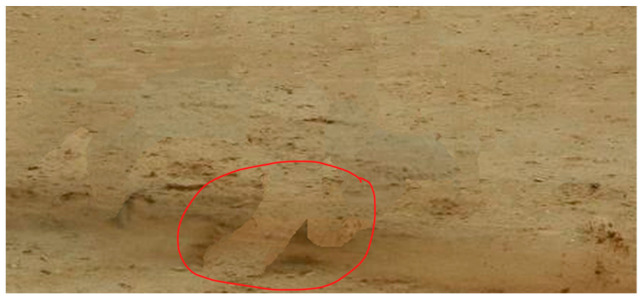
Sample artifact introduced by Gimp.

**Figure 25 jimaging-10-00042-f025:**
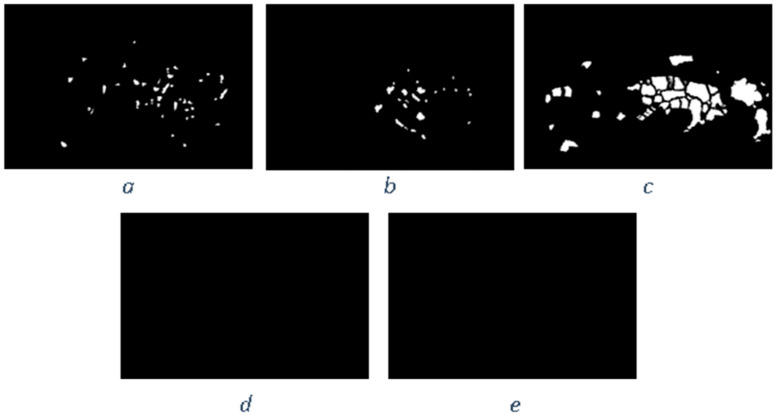
DEBI results on inpainted images using Criminisi (**a**), Gimp (**b**), Non-local patch (**c**), Lama (**d**), Mat (**e**).

**Figure 26 jimaging-10-00042-f026:**
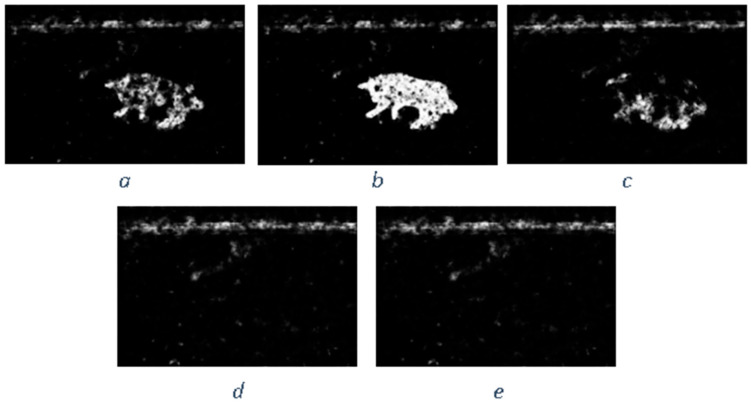
Mantranet results on inpainted images using Criminisi (**a**), Gimp (**b**), Non-local patch (**c**), Lama (**d**), Mat (**e**).

**Figure 27 jimaging-10-00042-f027:**
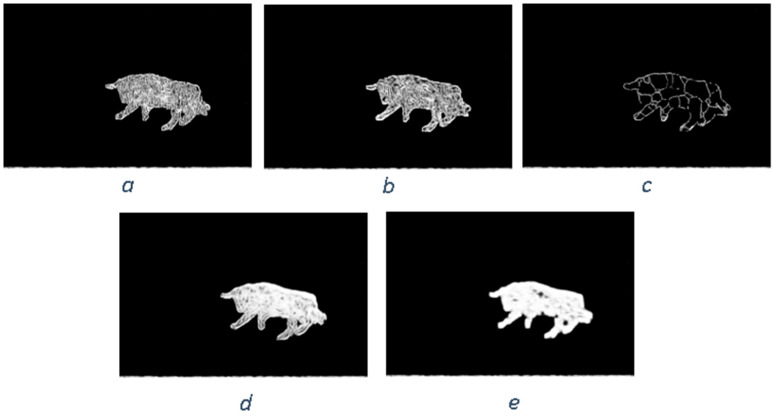
IID results on inpainted images using Criminisi (**a**), Gimp (**b**), Non-local patch (**c**), Lama (**d**), Mat (**e**).

**Figure 28 jimaging-10-00042-f028:**
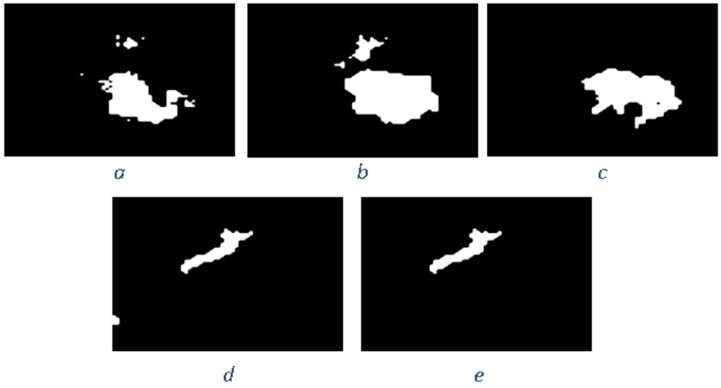
Focal results on inpainted images using Criminisi (**a**), Gimp (**b**), Non-local patch (**c**), Lama (**d**), Mat (**e**).

**Figure 29 jimaging-10-00042-f029:**
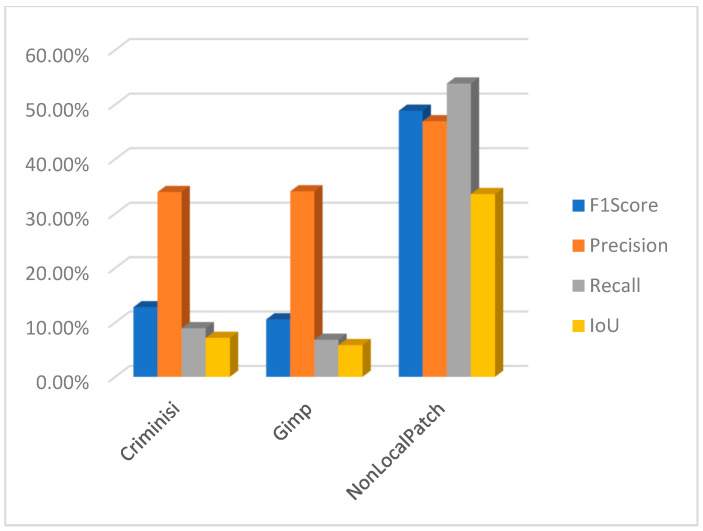
Evaluation metrics for the results of DEBI detection method applied on the inpainted Open Images Dataset V7 dataset with the following inpainting methods: Criminisi, Gimp, and Non-local Patch.

**Figure 30 jimaging-10-00042-f030:**
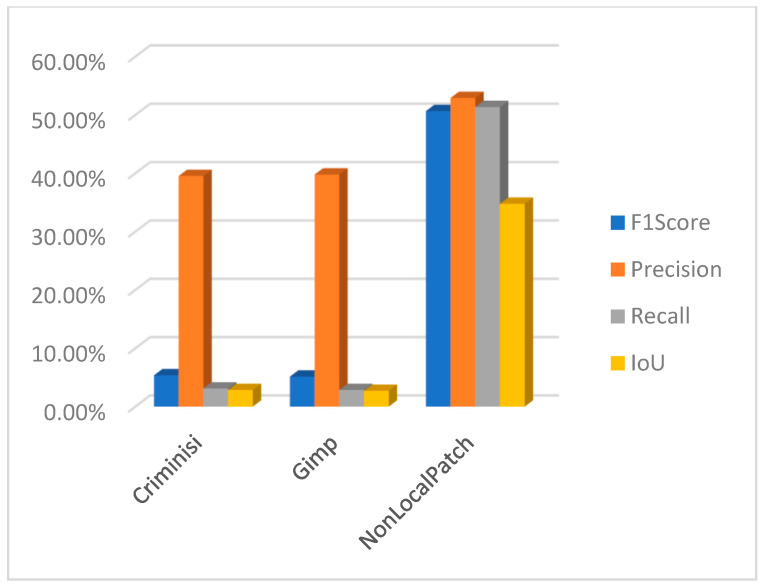
Evaluation metric for the results of CMFD detection method applied on the inpainted Open Images Dataset V7 dataset with the following inpainting methods: Criminisi, Gimp, and Non-local Patch.

**Figure 31 jimaging-10-00042-f031:**
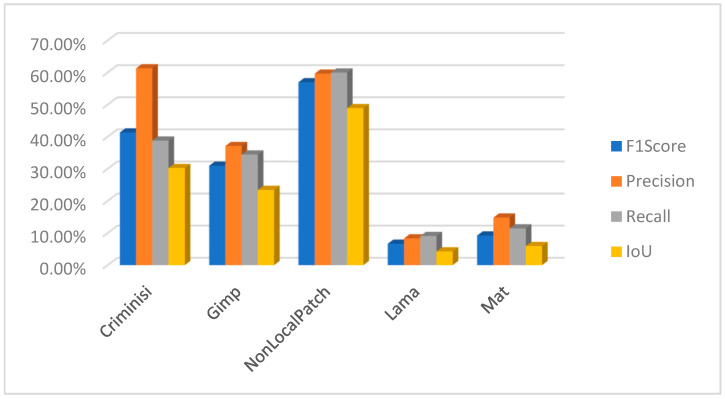
Evaluation metric for the results of focal detection method applied on the inpainted Open Images Dataset V7 dataset with the following inpainting methods: Criminisi, Gimp, Non-local Patch, Lama, and Mat.

**Figure 32 jimaging-10-00042-f032:**
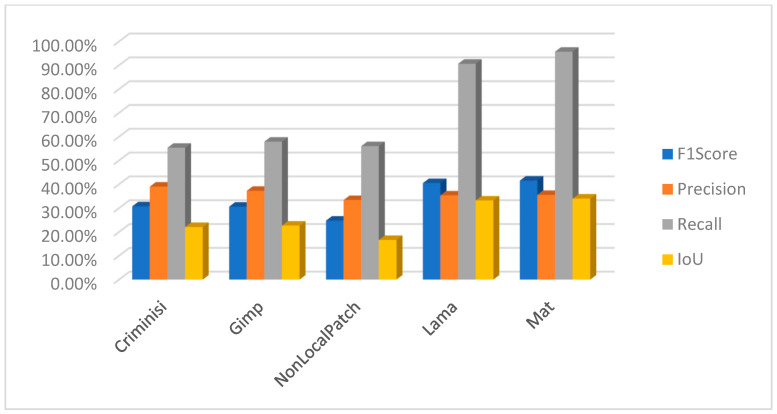
Evaluation metric for the results of IID-NET detection method applied on the inpainted Open Images Dataset V7 dataset with the following inpainting methods: Criminisi, Gimp, Non—local patch, Lama, and Mat.

**Figure 33 jimaging-10-00042-f033:**
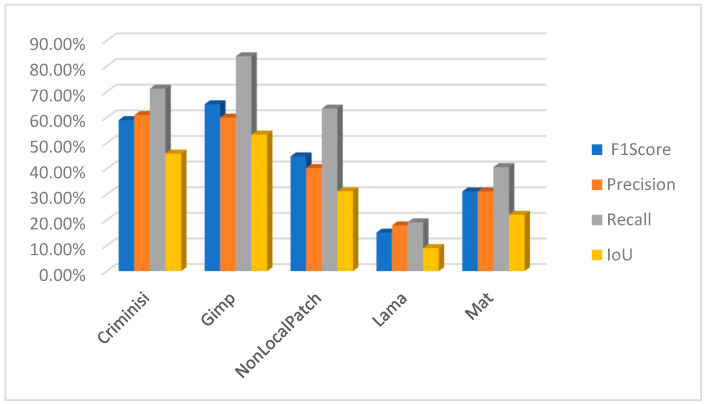
Evaluation metric for the results of Mantranet detection method applied on the inpainted Open Images Dataset V7 dataset with the following inpainting methods: Criminisi, Gimp, Non-local Patch, Lama, and Mat.

**Figure 34 jimaging-10-00042-f034:**
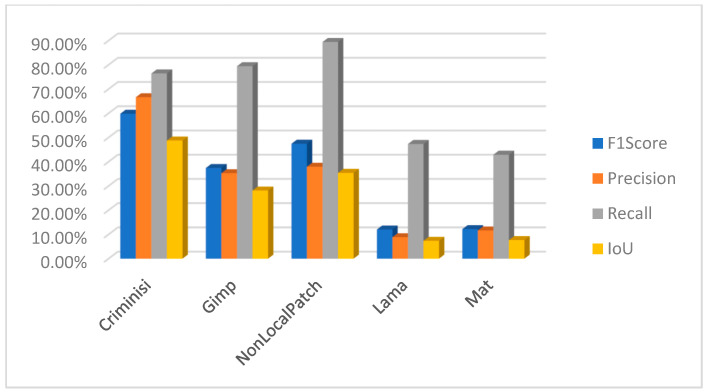
Evaluation metric for the results of PSCC-NET detection method applied on the inpainted Open Images Dataset V7 dataset with the following inpainting methods: Criminisi, Gimp, Non-local patch, Lama, and Mat.

**Figure 35 jimaging-10-00042-f035:**
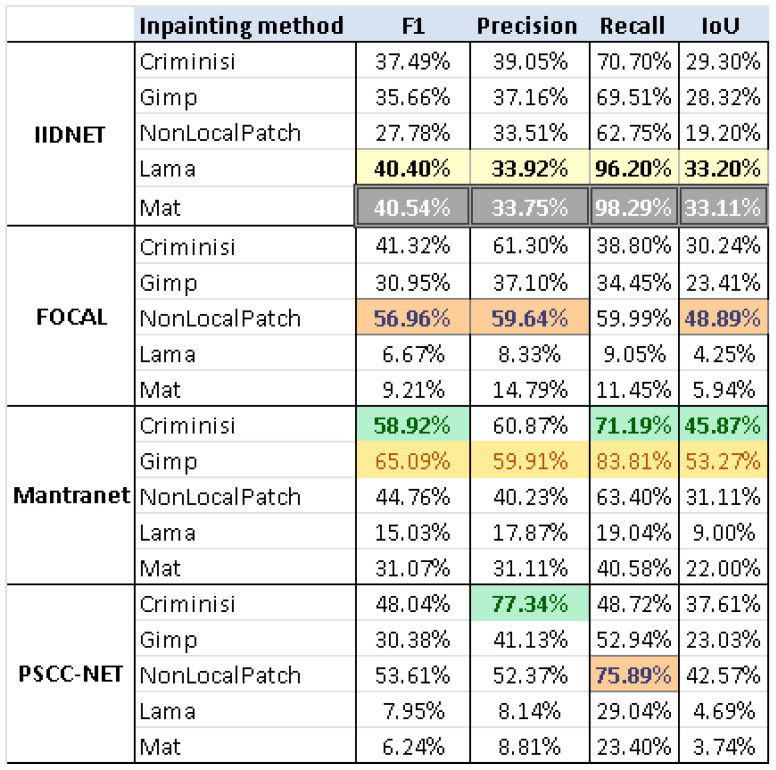
Summary of evaluation metrics for the results of IID-NET, focal, Mantranet, and PSCC-NET detection methods applied on the inpainted Open Images Dataset V7 dataset with the following inpainting methods: Criminisi, Gimp, Non-local patch, Lama and Mat.

**Table 1 jimaging-10-00042-t001:** Classic inpainting forgery detection.

Reference Article	Year	Observations
[59]	2008	The first found method that tackles inpainting methods. They evaluated against the Criminisi dataset. The method relies on detecting similar patches and applies fuzzy logic for similar patches.
[60]	2013	They continue the work in [63] and add several mechanisms to exclude a lot of false positives
[61]	2015	The authors produced two proposals—first do not compute the block differences and only compare the central pixel (this improves performance, and the accuracy is not that much affected); secondly, they proposed an improved method comparing to [64] of filtering and eliminating false positives.
[62]	2013	Similar method as the one in [64]. For better results/faster computation, they suggest a jump patch as the best approach.
[64]	2018	The same authors that proposed [65] included an additional step that consists of ensemble learning. They rely on a generalized Gaussian distribution between the DCT coefficients of various blocks
[65]	2015	The authors took the CMFD framework proposed in [62] but used feature extractions such as the Gabor magnitude
[66]	2018	The authors took the CMFD framework proposed in [62] but used the color correlations between patches for feature extraction
[67]	2020	The focus was on analyzing the reflectance of the forged and non-forged areas

**Table 2 jimaging-10-00042-t002:** Machine learning-based inpainting forgery detection.

Reference Article	Year	Observations
[72]	2017	The main idea was to use an SVM classifier composed of the following features: local binary pattern features, gray-level co-occurrence matrix features, and gradient features (actually they suggest using 14 features extracted from patches).
[73]	2018	Standard CNN model on which they trained original/altered patches.
[74]	2019	Same idea as [73], but they chose a Resnet model
[75]	2020	The authors employed a combination of Resnet and LSTM to better portray the differences between altered vs. non-altered regions. All the above methods were assessed against the initial Criminisi paper, and thus did not have to “compete” with latest image inpainting methods at the time.
[76]	2021	A tweaked version of a VGG model architecture
[77]	2022	A CNN model with a focus on detecting noise inconsistencies
[78]	2022	A U-NET VGG model that adds an enhancement block of five filters (four SRM + Laplacian) to be able to better detect inpainted areas.
[79]	2022	The authors suggest using three enhancements blocks: a steganalysis rich model to enhance noise inconsistencies, pre-filtering to enhance discrepancies in high-frequency components, and a Bayar filter to enable the adaptive acquisition of low-level prediction residual features like Mantranet architecture.

**Table 3 jimaging-10-00042-t003:** General forgery datasets for which a subset can be used for inpainting/object removal detection.

Name	Dataset Size (GB)	Number of Pristine/Forged Pictures	Image Size (s)	Type *	Mode	Observation
MICC	6.5 GB	1850/978	722 × 480 to 2048 × 1536	CP/OR	M	Some of the images are not very realistic, but it tries to generate several types of copy-moves by applying rotation and scaling. The problem is that the forged area is always rectangular
CMFD	1 GB	48/48	3264 × 2448 3888 × 2592 3072 × 2304 3039 × 2014	CP/OR	M	Very realistic dataset. Some of the images, since they use professional tools, are a mix of copy-move, object removal, and sampling. Grouped by camera type. The important thing to notice is that there are no post-processing operations performed on the images, but because of the high-quality/size, researchers can do their own post-processing. Images were processed using GIMP.
CoMoFoD	3 GB	200 + 60/ 200 + 60	512 × 512 3000 × 2000	CP/OR	M	Canon camera used only. They have used six post-processing operations—for e.g., JPEG compression with nine various quality levels or changing brightness, noise etc. The operations were performed in Photoshop. 3 GB is the only small variant of the dataset
CASIA	3 GB V2	7491/5123	160 × 240 to 900 × 600	CP + S	N	Contains different types of copied areas with resizing, rotation and post-processing of the forged area.
COVERAGE	150 MB	100/100	235 × 340 to 752 × 472	CP	N	Original images already contain similar objects, thus making them harder to detect. The forged is large—60% of the images have at least 10% forged area.
Realistic Tampering Dataset	1.5 GB	220/220	1920 × 1080	CP/OR	M/A	The dataset contains four different types of cameras and focuses on the inconsistencies at noise level between patches. The images were pre/post-processed with GIMP
MFC	150 GB	16 k/2 M	All sizes	CP/OR	N	They have used a series of techniques from simple copy-move to content-aware filling, seam carving, etc.

* Type CP = copy-move, OR = object removal, S = splicing; Mode M = manual, A = automatic, N = not mentioned.

**Table 4 jimaging-10-00042-t004:** Image inpainting forensic dataset.

Name	Dataset Size (GB)	Number of Pristine/Forged Pictures	Image Size (s)	Observation
DEFACTO INPAINTING	13 GB	10,312/25,000 (they have applied inpainting for same image but for different areas)	180 × 240 to 640 × 640	Some of the images are not very realistically inpainted due to the automatic randomized selection of the area from the MSCOCO dataset
IMD2020	38 GB	35 k/35 K	640 × 800 to 1024 × 920	Some of the images are not very realistic, and the forged images underwent some additional changes (some noise filtering/color uniformization). An interesting fact is that they manually selected areas and then used an automated algorithm, which means no post-processing/enhancements
IID-NET	1.2 GB	11 k/11 k	256 × 256	Random masks (based on MSCOCO) and 11 different automated algorithms for filling/removing objects The idea is interesting in terms of trying to tackle different inpainting algorithms, but still there were some problems in how the mask inpainted area is chosen. Also, another problem is that although several inpainting algorithms were tested, they were applied on different images.

## Data Availability

Once this paper will become available, we shall provide all the information, dataset, results and steps to reproduce. The details shall be available on this GitHub repository: https://github.com/jmaba/ImageInpaitingDetectionAReview.

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
