# Peer review of "Image Inpainting Forgery Detection: A Review"

_2313-433X, 2024, doi:10.3390/jimaging10020042_

Round 1

Reviewer 1 Report (Previous Reviewer 3)

Comments and Suggestions for Authors

My comments are well-addressed in general. Some minor checking on the syntax is recommended.

Author Response

We would like to thank you for your insightful and comprehensive review of our paper. Your valuable feedback has provided us with clear direction for improvement and has been instrumental in guiding our efforts.  

Reviewer 2 Report (Previous Reviewer 2)

Comments and Suggestions for Authors

The authors have revised the paper according to the comments. First, a questionable part was removed. Second, more elaborate review of the artifacts caused by inpainting was done. The paper can be published now.

Author Response

We would like to thank you for your insightful and comprehensive review of our paper. Your valuable feedback has provided us with clear direction for improvement and has been instrumental in guiding our efforts.  

Reviewer 3 Report (New Reviewer)

Comments and Suggestions for Authors

This paper provides insights into the research on inpainting methods and inpainting forgery detection.

However, the introduction predominantly focuses on the background of forgery detection rather than providing a comprehensive overview of inpainting. Specifically, Figure 2 and its corresponding content seem to deviate from the primary focus of the paper. The authors should consider emphasizing and elaborating more on image inpainting and inpainting forgery detection in the introduction.

Recent inpainting models have predominantly shifted towards deep learning approaches such as GANs or diffusion models, as opposed to diffusion-based and exemplar-based methods. However, this paper lacks substantial content on deep learning, with insufficient detailed explanations or comparisons of each technique, especially in Section 3.

The paper lacks a quantitative evaluation of inpainting techniques, and additional qualitative and quantitative experiments are necessary. Furthermore, as this is a review paper, it would be beneficial to conduct experiments with a wider range of models for a more comprehensive comparison in both quantitative and qualitative evaluations.

Comments on the Quality of English Language

The overall organization of the paper is not well-structured, leading to a significant decline in readability.

Author Response

We would like to express our gratitude for the perceptive review you provided on our paper. We greatly appreciate your comprehensive feedback, which has proven to be immensely valuable. We have thoroughly examined and implemented your suggestions into our work. Prior to addressing each of your criticisms, we would like to emphasize that this is the initial work of a PHD, and more papers focusing on methodologies and analysis in the field of inpainting detection will follow. We would want to address some considerations regarding your remarks:

  • the field of inpainting forgery detection is quite limited in scope. Until recently, most scholars have regarded it as a component of copy-move forgeries. However, it is evident, particularly with the most recent advancements in GAN or diffusion models, that this is not true. The inpainted region does not consist of directly replicated information from the surrounding surroundings, but instead, information is spread. Therefore, we find it important to discuss the category of forgery detection, specifically focusing on copy-move and rest approaches. This is necessary as it highlights the need to address inpainting methods separately, drawing inspiration from both copy-move and picture splicing detection techniques.
  • we somewhat agree with your second concern on deep learning approaches and are able to handle it. Nevertheless, when submitting this review, we carefully considered the previous feedback advising against excessive emphasis on the detailed analysis of inpainting techniques. Instead, we prioritized the examination of cutting-edge methods and dedicated more attention to the artifacts and other indicators generated by these methods, aiming to facilitate their detection. Consequently, we will incorporate an analysis of deep inpainting techniques, without altering the primary emphasis of the work. We would like to reiterate our profound gratitude for the time and effort you have dedicated to giving us such invaluable feedback. We appreciate your support and anticipate sharing our advancements with you soon.
  • the subsequent aspect we aim to discuss is the quantitative evaluation of inpainting. We would like to highlight that we have chosen two recent and highly influential deep learning inpainting papers that have shown promising results and have received significant citations. Our focus was on analyzing the artifacts and the outcomes of inpainting detection using these two methods.

    We have thoroughly evaluated all of your comments and have already commenced the process of executing the necessary modifications. We are fully committed to consistently improving our performance and are devoted to providing exceptional service that exceeds your expectations.
    We sincerely thank you for your time and effort in giving us invaluable input. We appreciate your support and anticipate sharing our advancements with you soon.

Round 2

Reviewer 3 Report (New Reviewer)

Comments and Suggestions for Authors

The authors have revised the paper according to the comments. While there are still minor points to be addressed, it appears to meet the standards required for current publication in its present form.

This manuscript is a resubmission of an earlier submission. The following is a list of the peer review reports and author responses from that submission.

Round 1

Reviewer 1 Report

Comments and Suggestions for Authors

While the paper and its attempted contribution surely has potentially some merits, I have several items of criticism that led me to reject the work.

I will go through the paper from first to last page, mentioning some main points of criticism or for improvement.

As a comment, a general flaw of the manuscript, making some parts difficult to read, is a missing substructure in terms of subparagraphs.

The introduction is supposed to contain not only an overview of what will be done, but also an examination of related work in the field. Here there is only one major reference, namely [5], and it is not clear why the authors' work is supposed to present an improvement over it.

I found the exposition of inpainting methods not too helpful in the context of the objective of the paper. For example, the authors conclude about the diffusion based methods that there are diffusion artefacts. This could have been concluded without giving a larger exposition. Also, it is not clear by the exposition how this looks like in a forgery, meaning how this could lead to detection. The exposition in this part as well as the other inpainting methods is in my opinion not technical enough to allow really to assess the artefacts, if the reader is not proficient with inpainting methods. On the other hand, for a review on inpainting by itself, also technical depth is missing. It is unclear for me why the authors describe algorithmic details about some methods, if this is superficial for detection. It seems that the reasoning is followed that in an overview part one should in good style write some sentences for each reference, but this misses in my opinion clearly the objective of the whole paper.

I would recommend to shorten the inpainting review considerably, mentioning for more information books like for example from Schoenlieb (a missing reference anyway) and e.g. from Weickert for the diffusion based methods. As a side note, I find strange that the method using Navier-Stokes methods is emphasized in this review, my impression of the field is that variational methods are much more adequate and prominent. Let me mention that I really did not like the exposition on diffusion based methods, also there are some mistakes there at the technical part (e.g. use of delta D). The exposition on exemplar based inpainting I found too focused on a certain selection of authors. This is surely an effect of the point that these sections make the attempt to give a review, but for that again it is too short.

Generally speaking, I do not see the benefit of the exposition of inpainting methods for this paper.

When doing an exposition on detection of inpainting artefacts, it would have been way better to show a few examples of obvious artefacts, using a few baseline methods. 

The exposition on inpainting forgery detection mechanisms, Section 3,  contains more interesting material, but exactly here when mentioning refs [3,5,55,56,57], there would have been the opportunity to discuss in detail what the submitted paper does different or better. This is still not done here.

Let me observe that I do not find the content of this section well structured. The  beginning and part 3.1. seems to contain some repetitions in content. The exposition in 3.1.1. gives an overview on technical steps, but it is unclear why the described algorithms may really work, meaning why it is possible to detect forgeries that way and why some of the described improvements tackle a flaw in the underlying basic scheme. This could have been illustrated nicely by some example. It is however clear that this cannot be done with all variations. Still, one could have discussed such methods in a more technical way when giving a review. For 3.1.2, again it is unclear what benefit the current paper has above refs [5,44]. The discussion of network architectures is unfortunately on a very shallow level.

I found the section on video object removal distraction from the main goal of the work.

The discussion about datasets is supposed to be one of the more interesting parts in the context of this work, but unfortunately here structure in content appears to be missing. To make this clear, a review work should contain an own attempt to give structure to the world of existing and previous work. Instead the authors just pile up one work after another. I acknowledge the more rich details here and the work behind, but as a review paper this is not good, there should be an attempt for organization.

In the discussion, of methods, I again find structure lacking. I have the criticism that the authors should discuss the organization of their dataset clearly and in higher detail, meaning the logic why they propose it that way and which methods should potentially perform better with what respect. When conducting the experiments, I find that interpretation of expectations of results missing clearly. The authors just take one method after the other and apply it, it would have been way better to show e.g. visual results for one experiment by all methods and discuss expected differences and qualities. 

Furthermore, the discussion of the employed metrics should be given at first here, and it should be clearly explained why exactly these measures are supposed to be meaningful. Some parts of the discussion seems to be again a repetition of what has been explained before. 

As a side note, if the underlying image is mainly dark / black, it is often helpful to invert colours for visualization.

In total, I have some appreciation of the work the authors conducted, but my general opinion is that significant work is still to be done before the paper has reached the level expected for publication in an international journal. Too much for a major revision. It is difficult to really say something about the expected contributions in detail before a major rewriting has been done that allows to assess the work much better. The paper has to be focused and more effort has to be put on the presentation in comparison and review part.

Comments on the Quality of English Language

There is a moderate (not small!) number of minor mistakes, please have another person at your institute check the mansucript after changes. I refrain from giving a list here.

Please avoid the usage of abbreviations like e.g. "we've" in a scientific paper.

I found very irritating how you employed hyphens, please change that.

Reviewer 2 Report

Comments and Suggestions for Authors

In general this is quality work containing much of the field in study. Systematization, analysis, and comparison are done. The language, structure, and logic are clear.

The problem is a bit obscure mathematical definitions in lines 123-131 and a page further. First, contains declaration "x:R^n -> R^m". Formally this means that there is a function x, which maps a point from R^n to point in R^m. Then, what does "I:x->I(x)" mean? What entity the function x is transformed to? This looks incomprehensible for me. Commonly accepted approach is to perceive image as a mapping function: "I:R^n -> R^m", x as an element in R^n space, I(x) as an element in R^m space, introduce set of all operable elements of R^n as X, so all x's under consideration belong to X, split X into two subsets X=K+U, pixels K where function I(x) is known and pixels U where it is unknown, etc. Frankly speaking, I see no need in this questionable mathematical formalization, since it is not used further.

Reviewer 3 Report

Comments and Suggestions for Authors

This paper mainly examines various techniques encompassing conventional texture synthesis methods, as well as those based on neural networks. Following are some comments:

- In the introduction, authors should mention the novelty of this article with respect to existing survey articles.

- More logical explanations should be added in the paper.

- the English might be polished at places.

- Authors should highlight the advantages and disadvantages of your methods they are referring.   - Why did the existing schemes fail? Does no study try to address this aspect before? If yes, this has to be mentioned.   - More results of prior works should also be reported in tabulated form with scientific comparison and explanation. 

Comments on the Quality of English Language

The English might be polished at places.